# The molecular cytoarchitecture of the adult mouse brain

Jonah Langlieb[1,7], Nina S. Sachdev[1,7], Karol S. Balderrama[1], Naeem M. Nadaf[1], Mukund Raj[1], Evan Murray[1], James T. Webber[1], Charles Vanderburg[1], Vahid Gazestani[1], Daniel Tward[2], Chris Mezias[3], Xu Li[3], Katelyn Flowers[1], Dylan M. Cable[1,4], Tabitha Norton[1], Partha Mitra[3], Fei Chen[1,5 ✉] & Evan Z. Macosko[1,6 ✉]

The function of the mammalian brain relies upon the specification and spatial positioning of diversely specialized cell types. Yet, the molecular identities of the cell types and their positions within individual anatomical structures remain incompletely known. To construct a comprehensive atlas of cell types in each brain structure, we paired high-throughput single-nucleus RNA sequencing with Slide-seq[1,2]—a recently developed spatial transcriptomics method with near-cellular resolution—across the entire mouse brain. Integration of these datasets revealed the cell type composition of each neuroanatomical structure. Cell type diversity was found to be remarkably high in the midbrain, hindbrain and hypothalamus, with most clusters requiring a combination of at least three discrete gene expression markers to uniquely define them. Using these data, we developed a framework for genetically accessing each cell type, comprehensively characterized neuropeptide and neurotransmitter signalling, elucidated region-specific specializations in activity-regulated gene expression and ascertained the heritability enrichment of neurological and psychiatric phenotypes. These data, available as an online resource (www.BrainCellData.org), should find diverse applications across neuroscience, including the construction of new genetic tools and the prioritization of specific cell types and circuits in the study of brain diseases.

The mammalian brain is composed of a remarkably diverse array of cell types that display high degrees of molecular, anatomical and physiological specialization. Although the precise number of distinct cell types present in the brain is unknown, the number is presumed to be in the thousands[3,4]. These cell types are the building blocks of hundreds of discrete neuroanatomical structures[5], each of which has a distinct role in brain function. Advances in the throughput of single-cell RNA-sequencing technology have enabled the generation of cell type inventories in many individual brain regions[6–15], as well as the construction of broader atlases that coarsely cover the nervous system[16,17]. Furthermore, the application of new spatial transcriptomics techniques to the brain has begun to illuminate the spatial organization of brain cell types[12,18–20]. However, a full inventory of cell types across the brain, with their cell bodies localized to specific neuroanatomical structures, does not yet exist.

## Transcriptional diversity and cell type representation across neuroanatomical structures

To comprehensively sample cell types across the brain, we used a recently developed pipeline for high-throughput single-nucleus RNA sequencing (snRNA-seq) that has high transcript capture efficiency and

nuclei recovery efficiency, as well as consistent performance across diverse brain regions[8,9]. We dissected and isolated single nuclei from 92 discrete anatomical locations derived from 55 individual mice (Fig. 1a, Methods and Supplementary Table 1). Across all 92 dissectates, after all quality control steps (Methods, Extended Data Fig. 1a and Supplementary Table 2), we recovered a total of 4,388,420 nuclei profiles with a median transcript capture of 4,884 unique molecular identifiers (UMIs) per profile (Extended Data Fig. 1b–e). We sampled nearly equal numbers of profiles from male and female donors, with minimal batch effects across mice, such that replicates of individual dissectates contributed to each cluster (Extended Data Fig. 1f). To discover cell types, we developed a simplified iterative clustering strategy in which the cells were repeatedly clustered on distinctions amongst a small set of highly variable genes until clusters no longer could be distinguished by at least three discrete markers (Methods). Our clustering algorithm largely recapitulated published results of the motor cortex[6] and cerebellum[9] (Extended Data Fig. 1g), and it was scalable to support the computational analysis of millions of cells (Methods). In total, after quality control, including doublet removal and cluster annotation (Methods), we identified 4,998 discrete clusters, the great majority of which (97%) were neuronal (Fig. 1a, Extended Data Fig. 1h and Supplementary Table 3), consistent with prior large-scale surveys of brain

[1]Broad Institute of Harvard and MIT, Cambridge, MA, USA. [2]Departments of Computational Medicine and Neurology, University of California, Los Angeles, Los Angeles, CA, USA. [3]Cold Spring Harbor Laboratory, Cold Spring Harbor, NY, USA. [4]Department of Electrical Engineering and Computer Science, Massachusetts Institute of Technology, Cambridge, MA, USA. [5]Harvard Stem Cell and Regenerative Biology, Cambridge, MA, USA. [6]Department of Psychiatry, Massachusetts General Hospital, Boston, MA, USA. [7]These authors contributed equally: Jonah Langlieb, Nina S. Sachdev. ✉e-mail: chenf@broadinstitute.org; emacosko@broadinstitute.org

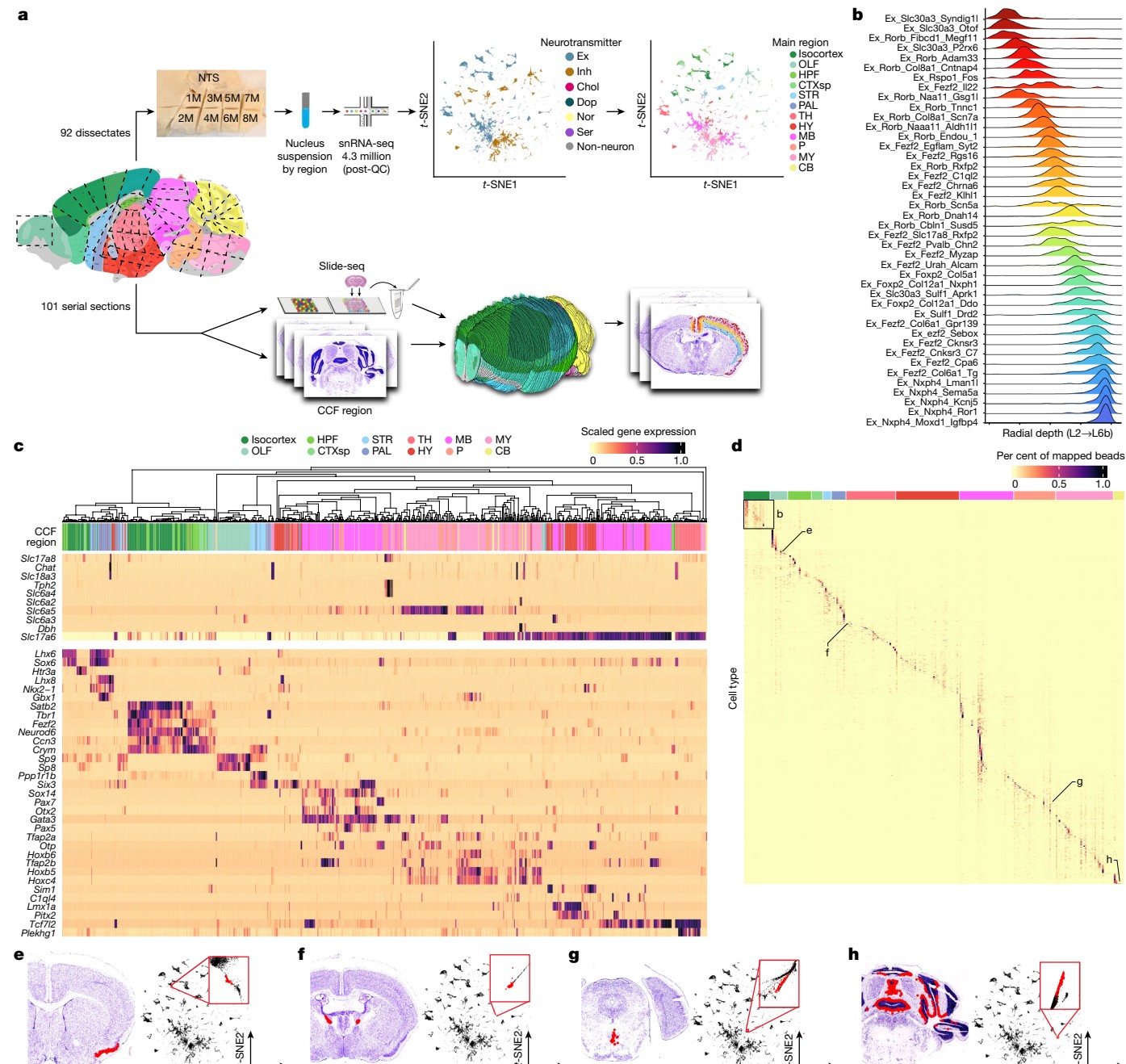

**Fig. 1 | Spatially mapping cell types using whole-brain snRNA-seq and Slide-seq datasets. a**, Schematic of the experimental and computational workflows both for whole-brain snRNA-seq sampling (upper arrows) and for Slide-seq sampling and CCF alignment (lower arrows). The *t*-distributed stochastic neighbour embedding (*t*-SNE) representations of gene expression relationships amongst 1.2 million spatially mapped snRNA-seq profiles (downsampled from 4.3 million) are coloured by neurotransmitter identity (upper left panel) and most common spatially mapped main region (upper right panel). Adapted from ref. 5, Allen Institute. **b**, Ridge plot depicting the spatial distributions of excitatory cortical cell types along the laminar depth of cortex (layers 2 to 6b) in the Slide-seq dataset. **c**, Heat maps depicting expression of the main neurotransmitter genes (upper panel) and canonical neuronal cell type markers (lower panel) across all 1,260 spatially mapped neuronal clusters. Cell types are annotated by the cluster dendrogram. **d**, Heat maps showing the spatial distributions of each spatially mapped cluster (rows)

within each DeepCCF structure (columns; a complete list is in Supplementary Table 4). Example mapped cell types in other panels are labelled on the heat map. **e**–**h**, Example confident mappings of neuronal cell types (confidence value > 0.3) (Methods) throughout the brain plotted in the CCF-aligned Slide-seq data (main plots) and in *t*-SNE space (insets) for the following cell types: Ex_Rorb_Ptpn20 (**e**, 35 arrays, 3,140 confident beads total), Ex_Ebf2_Iigp1_1 (**f**, two arrays, 84 confident beads total), SerEx_Fev_A2m (**g**, six arrays, 201 confident beads total), Inh_Nrk_Kctd16 (**h**, 25 arrays, 4,918 confident beads total). Scale bars, 1 mm. CB, cerebellum; CTXsp, cortical subplate; Chol, cholinergic neurons; Dop, dopaminergic neurons; Ex, excitatory neurons; HPF, hippocampal formation; HY, hypothalamus; Inh, inhibitory neurons; L, cortical layer; MB, midbrain; MY, medulla; NTS, nucleus tractus solitarii; Nor, noradrenergic neurons; OLF, olfactory areas; P, pons; PAL, pallidum; STR, striatum; Ser, serotonergic neurons; TH, thalamus; QC, quality control.

cell types[16,17]. Across the brain, we estimate that our sampling depth reached an estimated 90% saturation of cell type discovery (Methods and Extended Data Fig. 1i).

To determine the spatial distributions of these cell types, we next performed Slide-seq[1,2] on serial coronal sections of one hemisphere of an adult female mouse brain (Methods) spaced approximately 100 μm, matching the resolution of commonly used neuroanatomical atlases[21,22]. Slide-seq detects the expression of genes on 10-μm beads across the transcriptome within a fresh-frozen tissue section, providing near-cellular resolution data. In total, we sequenced 101 arrays, spanning the entire anterior–posterior axis of the brain. We aligned the sequencing-generated Slide-seq images to images of adjacent histological sections, which are rich in neuroanatomical detail. To assign beads to specific neuroanatomical atlas structures, we aligned the adjacent histological sections to the Allen Common Coordinate Framework[5] (CCF) (Methods and Extended Data Fig. 2a). This CCF provides hierarchical regional definitions, allowing us to tag each Slide-seq bead with a 'Main Region'—1 of 12 large structural components of the brain (enumerated in Fig. 1a)—as well as more fine-grained regional definitions, which we call 'DeepCCF' structures (listed in Supplementary Table 4). To confirm the accuracy of our alignment, we plotted the expression of three highly region-specific markers across our CCF-defined regions and quantified the distance of each expressing bead from the expected CCF region (Extended Data Fig. 2b). From this analysis, we estimate our alignment error to be in the range of 22–94 μm (Extended Data Fig. 2c).

To localize cell types to brain structures, we computationally decomposed individual Slide-seq beads into combinations of snRNA-seq-defined cluster signatures using Robust Decomposition of Cell Type Mixtures (RCTD)[23]. To handle the enormous cellular complexity of these regions, we implemented RCTD in a highly parallelized computational environment[24] and developed a confidence score that more accurately distinguishes among groups of highly similar cell type definitions (Methods). In total, we mapped 1,937 snRNA-seq-defined clusters (Methods) to greater than 1.7 million beads within the Slide-seq dataset. We computed the cortical layer depth of a set of 42 isocortical excitatory neuronal types and found that the mappings had the expected highly regionalized radial depth[7] (Fig. 1b) when ordered by their best integrated match with a previous cortical atlas[7], suggesting faithful projection of cell type signatures into spatial coordinates.

Most glial populations were distributed across large neuroanatomical boundaries (telencephalon, mesencephalon and rhombencephalon), indicating that, relative to neurons, regional gene expression differences amongst glial populations were small (Extended Data Fig. 2d). A single oligodendrocyte precursor cluster was identified, in contrast to a recent report of additional oligodendrocyte precursor subspecialization in humans[25]. The glial clusters with regional segmentation included astrocytes, which divided into olfactory-specific, telencephalic and non-telencephalic populations, as well as a cerebellum-specific population (the Bergmann glia). Amongst our endothelial cell populations, we identified populations preferentially localized to the choroid plexus (Extended Data Fig. 2e). Additional regionally localized glial populations included the olfactory ensheathing neurons, identified by their expression of the known marker homeobox genes *Alx3* and *Alx4* (ref. 26), and hypothalamic tanycytes, which uniquely express *Rax*[27].

To facilitate interpretation and visualization of these large numbers of neuronal populations, we performed hierarchical clustering, plotting known markers of cell type identity across the leaves of the dendrogram (Methods). We assessed the consistency of expression of these known markers (mostly transcription factors) with the expected localizations of cell types across 12 main brain regions defined in the Allen Brain Atlas (Fig. 1c): isocortex, the olfactory areas, hippocampal formation, striatum, pallidum, hypothalamus, thalamus, midbrain, pons, medulla and cerebellum. Amongst our neuronal clusters, we identified cortical, amygdalar, olfactory and hippocampal excitatory projection neurons (*Tbr1*, *Neurod6* and *Satb2*); telencephalic interneurons (*Sp8*, *Sp9* and

*Htr3a*); spiny projection neurons (SPN) of the striatum and adjacent pallidal structures (*Ppp1r1b*); hypothalamic neurons (*Nkx2-1*, *Sim1*, *Lhx6* and *Lhx8*); principal neurons of the thalamus (*Tcf7l2*, *Six3* and *Plekhg1*); neurons of the brain stem that populate mostly midbrain and pontine structures (*Otx2*, *Gata3*, *Pax5*, *Pax7* and *Sox14*); neurons expressing Hox homeobox genes that are primarily in the rhombencephalon; and cerebellar neurons expressing *Tfap2a* and *Tfap2b*. Neurons also specialize in the specific neurotransmitters they express. We detected discrete populations of gluatmatergic (*Slc17a6*, *Slc17a7* and *Slc17a8*), γ-aminobutyric acid (GABA)-ergic (*Slc32a1*), glycinergic (*Slc6a5*), cholinergic (*Chat* and *Slc18a3*), serotonergic (*Slc6a4* and *Tph2*), dopaminergic (*Slc6a3*) and noradrenergic (*Slc6a2* and *Dbh*) cell types distributed in the expected regions. By combining knowledge of marker expression patterns with spatial localization of cell types, we annotated the neuronal clusters of the dendrogram into a smaller set of 223 metaclusters (Supplementary Table 5), many of which corresponded to known, named cell types within the various structures of the brain (Supplementary Table 6). Together, these results indicate that our systematic sampling covered the expected molecular diversity of neurons across the mouse brain.

Most neuronal populations were mapped to specific and neuroanatomically related structures (Fig. 1b,d–h), reflecting the strong regional specificity of neuronal specializations. We assessed the distribution of neuronal cell types within DeepCCF structures. Most cell types showed highly refined regional localization; 60% of mapped clusters were confidently mapped (Methods) to three or fewer DeepCCF regions, reflecting the extent to which neuroanatomical nuclei are individually composed of locally diversified cell types.

## Variation in neuronal diversity across neuroanatomical structures

Our initial results revealed surprisingly large numbers of cell types distributed across the main brain regions. To explore cellular diversity at a finer neuroanatomical scale, we tallied the number of cell types confidently mapping to each DeepCCF structure, computing the number of types needed to occupy 95% of all mapped beads localized within that DeepCCF structure (Methods). Within the 12 main brain regions, we found the largest diversity of cell types in the midbrain, followed by hypothalamus, pons and medulla (Fig. 2a). Within the more fine-grained DeepCCF structures, we found particularly high cell type diversity within the periaqueductal grey matter and reticular nucleus of the midbrain. Regions of high diversity in other major brain areas included the parvicellular reticular nucleus of the medulla, the pontine reticular nucleus, the lateral hypothalamic area and the bed nucleus of the stria terminalis, consistent with our prior analysis of this area[8]. Although cell types were often highly focal within DeepCCF structures (Fig. 1b,d–h), some cell types also crossed DeepCCF boundaries. To visualize cellular compositional relationships amongst brain regions in greater detail, we built a force-directed graph in which the edges between DeepCCF regions were weighted to represent the number of clusters that jointly mapped in those regions (Methods and Fig. 2b). Cell types largely were restricted to each major brain area but showed greater mixing between pons and medulla compared to other regions, indicating more mixing of cell types specifically within those structures (Extended Data Fig. 3a).

Circuit-level analyses of the mouse brain have relied upon the availability of genetically delivered molecular tools to excite, inhibit and record from individual neuronal populations. These tools have historically been delivered to specific subpopulations of neurons through the use of recombinase-based systems, but more recently, RNA editing-based strategies have been developed to enable translation of transgenes only in the presence of specific endogenous messenger RNA transcripts[28–30]. Both strategies require nominating small numbers of high-value marker genes that can optimally distinguish amongst many

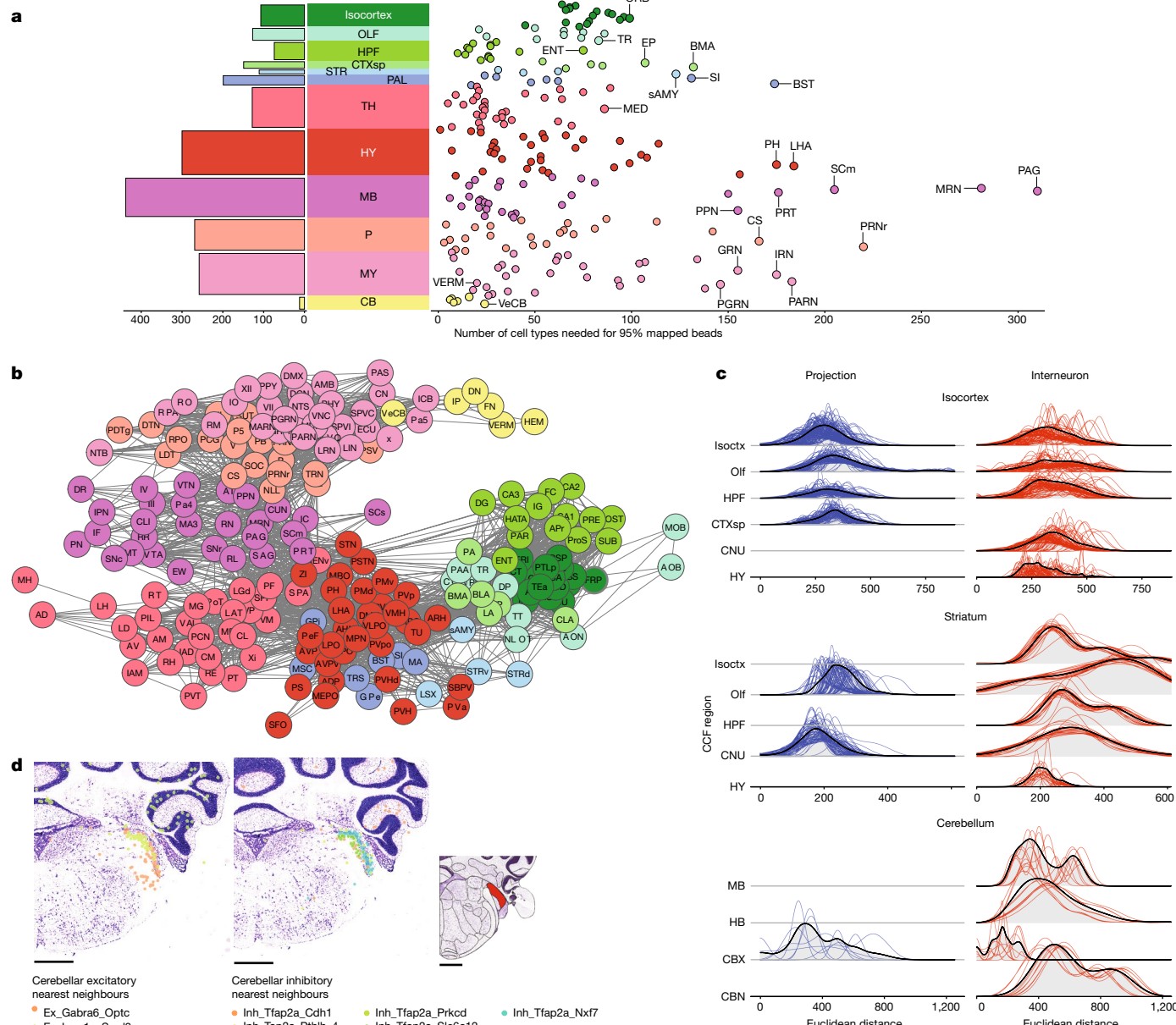

**Fig. 2 | Variation in neuronal diversity across neuroanatomical structures.**
**a**, Cumulative number of cell types needed to reach 95% of mapped beads in each DeepCCF region (right panel; defined in Supplementary Table 4) and averaged within individual main regions (left panel). The DeepCCF regions with the largest values are labelled. **b**, Force-directed graph showing cell type sharing relationships amongst DeepCCF regions. Edges are weighted by the Jaccard overlap between each region (Methods). **c**, Ridge plot depicting the transcriptomic distance from selected regions' projection and interneuron cell types to their proximate neighbourhoods, separating each neighbourhood into their CCF regions (Methods). **d**, Example confident mappings of the nearest neighbour cell types (confidence value > 0.3) (Methods) for cerebellar interneurons grouped by excitatory and inhibitory index cell types. Inset highlights the neuroanatomical annotation of the dorsal cochlear nucleus from the Allen Mouse Reference Atlas. Adapted from ref.5, Allen Institute. Scale bars, 1 mm. CBN, cerebellar nuclei; CBX, cerebellar cortex; CNU, cerebral nuclei; HB, hindbrain; Isoctx, isocortex.

distinct clusters. To identify the minimum number of genes needed to combinatorially define each cell type in our snRNA-seq dataset, we framed the question as a set cover problem[31] (Methods and Supplementary Methods), which can be solved to optimality using mixed integer linear programming techniques[32,33]. Our algorithm effectively identified a minimally sized set of defining genes for a great majority of cell types (93%), requiring a median of three genes (Extended Data Fig. 3b; all combinations are detailed in Supplementary Table 7).

When we performed the analysis on each of the 12 major brain regions separately, twice as many cell types could be uniquely defined by up to two genes (Extended Data Fig. 3c). The minimally defining genes were enriched for transcription factors (odds ratio = 2.54, $P < 0.001$),

G-protein coupled receptors (GPCRs; odds ratio = 1.83, $P < 0.001$) and neuropeptides (NPs; odds ratio = 5.76, $P < 0.001$) (Methods and Extended Data Fig. 3d), gene families that have been historically used to define cell types in the brain.

Similar cell types are known to populate different brain areas. For example, inhibitory neurons derived from the medial ganglionic eminence and caudal ganglionic eminence are found throughout telencephalic structures, such as the striatum, amygdala, hippocampus and isocortex. In our neuronal dendrogram, we had identified metaclusters, which included cortical medial ganglionic eminence-derived and caudal ganglionic eminence-derived neurons, based upon their isocortical localizations, as well as expression of key lineage markers,

such as *Lhx6*, *Nkx2-1* and *Sp8* (Supplementary Table 6). For each of these neuronal cell types, we defined a neighbourhood of clusters in close proximity within the dendrogram and examined their relative spatial distributions across brain areas (Methods and Extended Data Fig. 4). Interestingly, molecular relatives of these inhibitory neurons were found throughout the telencephalon—including in striatal and pallidal structures—as well as in the hypothalamus. By contrast, using the same neighbourhood definition for excitatory isocortical neurons—which are the long-range projection neurons of the cortex—revealed cell types with a more limited distribution, only within other cortical structures like the hippocampus and olfactory cortex (Fig. 2c).

We wondered whether the above result—observing more spatially restricted molecular specialization amongst projection neurons compared with local interneurons—might be more generally observed throughout the brain. We therefore repeated the same analysis on two other brain areas for which the projection versus interneuron distinctions amongst transcriptionally defined cell types are well known: the striatum and cerebellar cortex. Examination of the neighbourhoods of cell types in the striatum revealed the same pattern, in which the spiny projection neurons showed close cellular relatives within only pallidal and striatal structures, whereas the interneuron populations had relatives spread throughout the telencephalon (Fig. 2c). Similarly, in the cerebellum, the projection neurons—Purkinje cells—had no molecularly similar relatives outside the cerebellar cortex, whereas the cerebellar interneurons had close relatives in several brain stem structures, such as the dorsal cochlear nucleus (Fig. 2d). Together, these results suggest that regional specialization in the brain is strongest in the principal projection neurons of individual structures, whereas interneurons are more likely to retain molecular features that are shared across different brain areas.

## Principles of neurotransmission and NP usage

Neurons communicate with each other across synapses through the expression of different small molecules and peptides. We asked in which regions and in which combinations neurotransmitters are used across the cell types of the brain. Because the production and usage of these neurotransmitters at synapses require different sets of gene products, we leveraged our snRNA-seq data to assign neurotransmitter identities to each cell type (Methods).

Overall, amongst the neuronal snRNA-seq clusters, cell type diversity was well balanced between excitatory and inhibitory cell types (2,420 excitatory and 2,246 inhibitory), and co-transmission of glutamate with an inhibitory neurotransmitter (GABA or glycine) was relatively rare (1.1% of all neuronal clusters) (Fig. 3a). Most co-expressing populations (35 of 54) expressed the glutamate transporter *Slc17a8* (VGLUT3) and derived from a wide range of lineages, populating regions across the telencephalon, midbrain and hindbrain. Amongst neuron types expressing neuromodulators, we found that the cholinergic neurons were more diverse (102 clusters) compared to serotonergic and dopaminergic types (25 and 13 clusters, respectively) and were distributed much more widely across the nervous system (Extended Data Fig. 5a,b).

Although the brain-wide cellular composition was balanced between inhibitory and excitatory types, individual brain regions are known to be composed of more skewed compositions of excitatory or inhibitory neurons. To characterize neurotransmission balance comprehensively in all structures, we quantified the excitatory-to-inhibitory balance of each DeepCCF region by comparing the ratio of the number of beads mapping to excitatory cell types with those mapping to inhibitory cell types (Methods). The computed excitatory-to-inhibitory balances recovered the expected broad patterns, including the dominance of excitatory cells in thalamic nuclei, and the lack of excitatory populations within the striatum (Fig. 3b). Furthermore, more subtle distinctions could also be appreciated, such as the higher inhibitory proportion in certain thalamic nuclei known to contain interneurons

(for example, LGd, the dorsal part of the lateral geniculate complex). Within the telencephalon, regions were more commonly skewed toward a predominantly excitatory (for example, cortical regions) or predominantly inhibitory (for example, striatum) composition. In addition, regions with high excitatory-to-inhibitory imbalance were more likely to be predominantly excitatory, whereas predominantly inhibitory regions were less common, being largely restricted to the striatum, the thalamic reticular nucleus and a few brain stem nuclei.

NPs exert varied and complex neuromodulatory effects on circuits through downstream GPCRs. NPs are also often co-expressed with other neurotransmitters to directly modulate synaptic activity. We utilized our spatially mapped cell type inventory to characterize the basic rules and principles by which NPs are used throughout the brain. We curated a set of 65 genes that produce at least one NP with a known downstream GPCR (Supplementary Table 8) and quantified the number of NP-expressing and GPCR-expressing cell types. Amongst our 4,998 cell types, 80.9% expressed at least one NP, underscoring the ubiquity of NP signalling in the mammalian central nervous system (Fig. 3c). Receptor expression was even more ubiquitous: 91.6% of cell types expressed receptors for more than three NPs. Historically, NP signalling has been particularly strongly associated with the hypothalamus, where many of the NPs were originally biochemically discovered[34]. However, our analyses did not find that, overall, hypothalamic neurons were any more likely to express NPs compared with neurons in other brain areas (Extended Data Fig. 5c). Rather, the hypothalamus, as well as the pallidum and midbrain, were more likely to express a subset of NPs—like oxytocin or vasopressin—that are highly selectively expressed, whereas other brain regions expressed NPs that were more ubiquitous throughout the nervous system (Fig. 3d).

Nearly all NPs and receptors were expressed by neuronal cell types (Fig. 3e). However, we identified two likely examples of NP signalling between neurons and glia. The expression of *Cartpt* was detected in 232 neuronal populations distributed in hypothalamic and midbrain regions, whereas its receptor *Gpr160* (ref. 35) was highly restricted to microglia and macrophage populations. Interestingly, *Gpr160* induction was observed to be within microglia in a recent study of spinal cord nerve injury[35]. Conversely, the expression of the angiotensin-encoding gene *Agt* was found to be primarily in astrocytes found in non-telencephalic regions (Fig. 3e and Extended Data Fig. 5d), whereas its receptors *Agtr1a* and *Agtr2* were enriched in non-telencephalic neurons. Astrocyte–neuron signalling through angiotensin could have important homoeostatic roles, particularly in the midbrain where dopaminergic neurons vulnerable to neurodegeneration in Parkinson's disease were recently identified to selectively express *Agtr1a*[36], and inhibition of the angiotensin receptor has been shown to be neuroprotective in Parkinson's disease animal models[37] and in clinical cohorts[38].

## Activity-dependent gene enrichment across cell types and regions

Neuronal cells, in response to an increase in action potential firing, induce the expression of hundreds of activity-regulated genes (ARGs)[39]. The prototypical ARG is *Fos*, which is induced within minutes of elevated activity, along with several highly correlated genes, including *Junb* and *Egr1*, which are collectively referred to as immediate early genes (IEGs). These IEGs have been primarily discovered and studied in excitatory cortical or hippocampal cells. Our Slide-seq and snRNA-seq atlases provide two key advantages for assessing ARG heterogeneity across cell types. First, they are comprehensive in their coverage of the brain to enable broad comparative analysis. Second, they are performed on brain tissue that is frozen immediately after animal perfusion, eliminating any post-mortem effects on ARG expression[40,41].

To characterize ARGs across neuronal types, we first partitioned our mapped clusters into 28 cell type groups defined by their Slide-seq

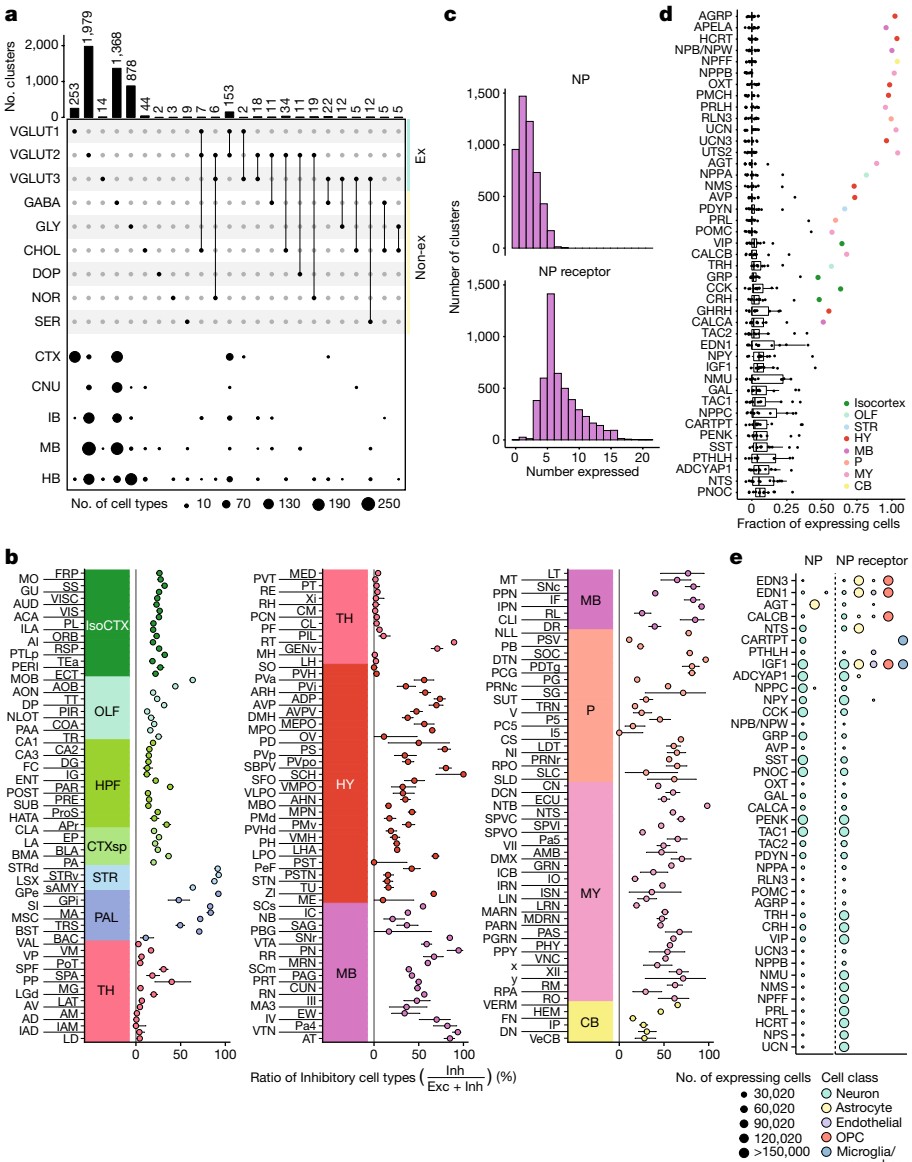

**Fig. 3 | Neurotransmission and NP usage across regions of the mouse brain.**
**a**, Upset plot of the frequency of neurotransmitter usage by individual snRNA-seq-defined cell types (upper panel). Dot plot depicting the spatial distribution of cell types in each of the neurotransmitter groups across major brain areas (lower panel). IB, interbrain. **b**, Point estimates of the fraction of each DeepCCF region composed of mapped inhibitory cell types. Data are presented as this calculated proportion (central dots) with the 95% confidence interval of the corresponding binomial distribution denoted by the error bars (Methods). **c**, Histograms denoting the number of distinct NPs (upper panel) and neuropeptide receptors (NPRs; lower panel) expressed in each

snRNA-seq-defined neuronal cell type. **d**, Fraction of all cells expressing each NP (y axis) in each of the 12 main brain areas. Regions accounting for more than 50% of total expression of that NP are coloured and labelled. n = 43 NPs examined over 1,182 cell types. Box plots are centred at the median and bounded by the interquartile range (IQR; 25th–75th percentiles), with the lower whisker at the data point greater than or equal to (25th percentile − 1.5 × IQR) and the upper whisker at the data point less than or equal to (75th percentile + 1.5 × IQR). **e**, Dot plot depicting the number of cells expressing each NP (left of the dotted line) and NPR (right of the dotted line) within each major cell class. OPC, oligodendrocyte precursor.

mapped region and their neurotransmitter identity (Methods). We then selected 406 candidate ARGs whose correlation with *Fos* was at least 0.3, met statistical significance (adjusted *P* < 0.05) and for which *Fos* was also above the 99.5% quantile of all correlations in at least one cell type group (Methods). To ensure robustness, we validated that our candidate ARGs were similarly correlated with another canonical IEG, *Junb* (Extended Data Fig. 6a). To identify which genes are consistently correlated across cell type groups, we constructed a bipartite graph, connecting each gene to cell type groups within which it is highly correlated with *Fos* (Methods). Examination of this graph revealed that the most connected genes—those that are most consistently and highly correlated with *Fos* across the brain—included most canonical IEGs,

such as *Egr1*, *Npas4*, *Arc*, *Junb*, *Btg2* and *Nr4a1*. We selected the eight most correlated of these genes to compare their relative activity across regions and cell types (Methods and Extended Data Fig. 6b). Expression of these IEGs across each region in our Slide-seq dataset was highest in the isocortex, olfactory bulb, striatum and amygdala, whereas regions of cerebellum and medulla showed the lowest average IEG expression (Fig. 4a and Extended Data Fig. 6c). Similarly, in our snRNA-seq clusters, IEG activity was noticeably higher in excitatory populations, particularly those in the isocortex, olfactory areas and hippocampal formation (Extended Data Fig. 6d).

Our candidate ARG set also contained many genes connected to only a few of the major cell type groups, suggesting heterogeneity

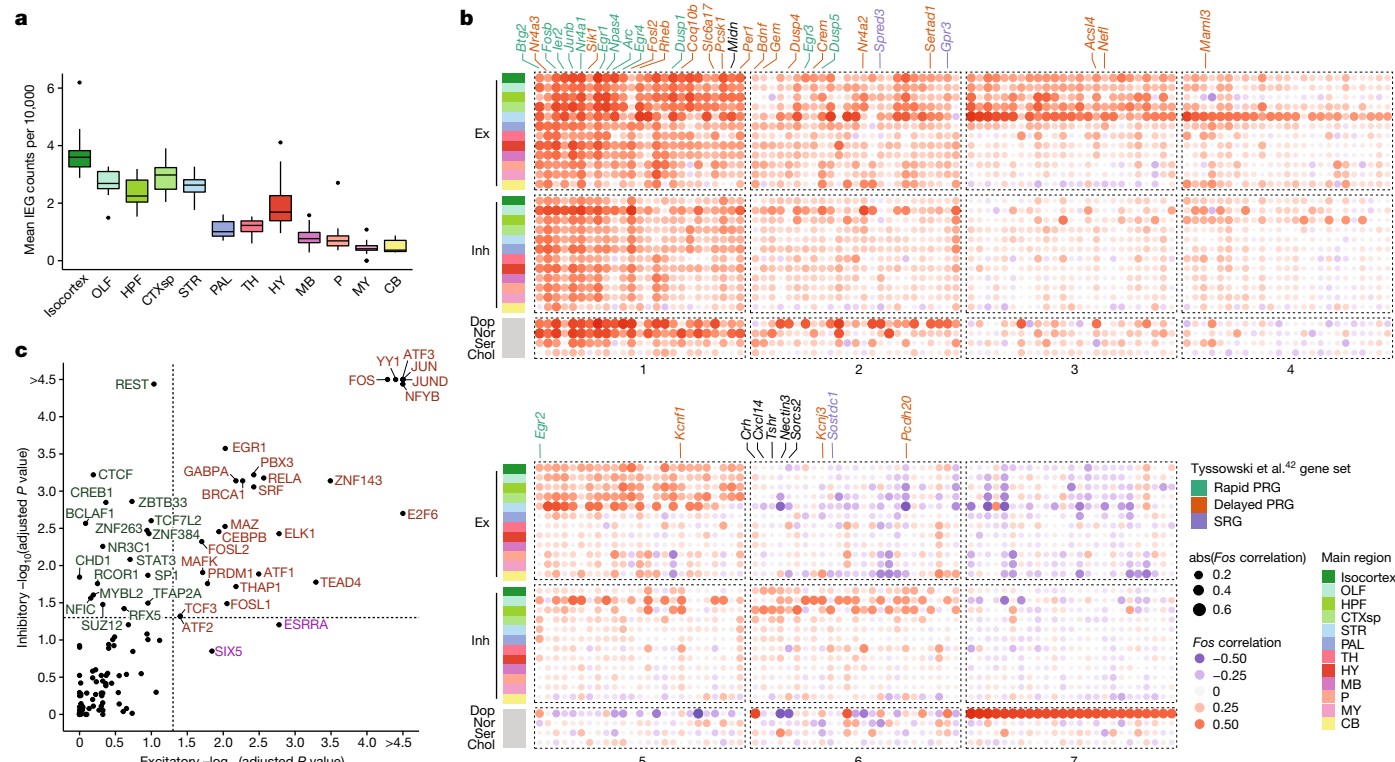

**Fig. 4 | Patterns of activity-dependent gene expression across brain regions. a**, Box plots quantifying mean core IEG Slide-seq counts per 10,000 coloured by main brain regions. $n = 232$ DeepCCF regions examined across 12 main regions. Box plots are centred at the median and bounded by the IQR (25th–75th percentiles), with the lower whisker at the data point greater than or equal to (25th percentile − 1.5 × IQR) and the upper whisker at the data point less than or equal to (75th percentile + 1.5 × IQR). **b**, Downsampled dot plot of correlation coefficients between *Fos* and candidate ARGs (columns) across major regions of the brain (rows). Genes are coloured by their established ARG gene set[42] identity if applicable. Numbers at the bottom correspond to ARG cluster identities as determined by hierarchical clustering. PRG, primary response gene; SRG, secondary response gene. **c**, Scatterplot quantifying transcription factor enrichment ($P < 0.05$, FDR corrected) between excitatory and inhibitory populations. Enrichment scores are computed by fgsea using a positive one-tailed test. Transcription factors are coloured by their cell type enrichment specificity.

in the transcriptional programs of cell types in response to activity. To more deeply explore cell type-specific ARGs, we hierarchically clustered our gene set into seven clusters. Clusters 1–4 were the most universally correlated across cell types and regions (Fig. 4b, Extended Data Fig. 6e and Supplementary Table 9) and were highly enriched for known ARGs[42] (Methods and Extended Data Fig. 6f). Cluster 1 also included *Midn*, recently discovered to have a key role in IEG protein stability[43]. Clusters 5–7, meanwhile, were more cell type specific; cluster 5 was relatively specific for telencephalic excitatory neurons, cluster 6 was more specific for telencephalic inhibitory neurons, and cluster 7 was specific for dopaminergic neurons. Our inhibitory-specific cluster 6 included several genes previously reported as activity regulated in cortical interneurons, such as *Crh* and *Cxcl14* (ref. 41). Many of these genes are implicated in dendritic spine development and re-modelling, such as *Tshr*[44], *Nectin3* (ref. 45) and *Sorcs2* (ref. 46), indicating that synaptic plasticity may be a particularly prominent component of the activity-related response in telencephalic inhibitory cell types. To explore how the transcription of these gene sets may be differentially regulated across cell types, we compared the enrichment of transcription factor targets between genes highly correlated with *Fos* in either telencephalic excitatory or inhibitory cells (Methods). Amongst the 46 transcription factors with significant enrichment ($P < 0.05$, false discovery rate (FDR) corrected) (Fig. 4c), most (26 transcription factors) were jointly enriched in both inhibitory and excitatory populations, but inhibitory cells were selectively enriched for the targets of 18 transcription factors. These transcription factors included several well-known chromatin re-organizers, including CTCF, BCLAF1, and CHD1, suggesting an important role for epigenetic modification of

inhibitory neurons in activity-dependent processes. Together, these analyses reveal how brain-wide, unbiased sampling of cell types can reveal not only the molecular markers defining these types but also conserved, dynamic patterns of gene regulation that occur across cell type groups.

## Heritability enrichment of neurological and psychiatric traits

Over the past 10 years, genome-wide association studies (GWAS) have uncovered risk loci associated with numerous neuropsychiatric traits. Identifying the cell types and brain regions in which these loci influence disease risk could catalyse new directions in understanding pathogenic mechanisms of many difficult-to-treat brain diseases. Because of their comprehensive coverage, our combined spatial and single-nucleus transcriptomics datasets provide a unique opportunity to investigate the relative enrichment of disease risk alleles across the entire mammalian nervous system. Several studies have integrated single cell and GWAS by aggregating cells from the same type and computing an enrichment statistic between the gene expression pattern of the cell type and the genes associated with risk by GWAS[11,36,47–49]. We used a recently described approach specifically developed for single-cell datasets[50] (Methods) to evaluate the relative enrichment of loci from 16 neurological and psychiatric traits across our spatially localized cell types (Supplementary Table 10).

After multiple hypothesis correction testing (Methods), we identified a total of 145 cell types across 11 traits that met statistical significance (adjusted $P < 0.05$) (Fig. 5a and Supplementary Table 11).

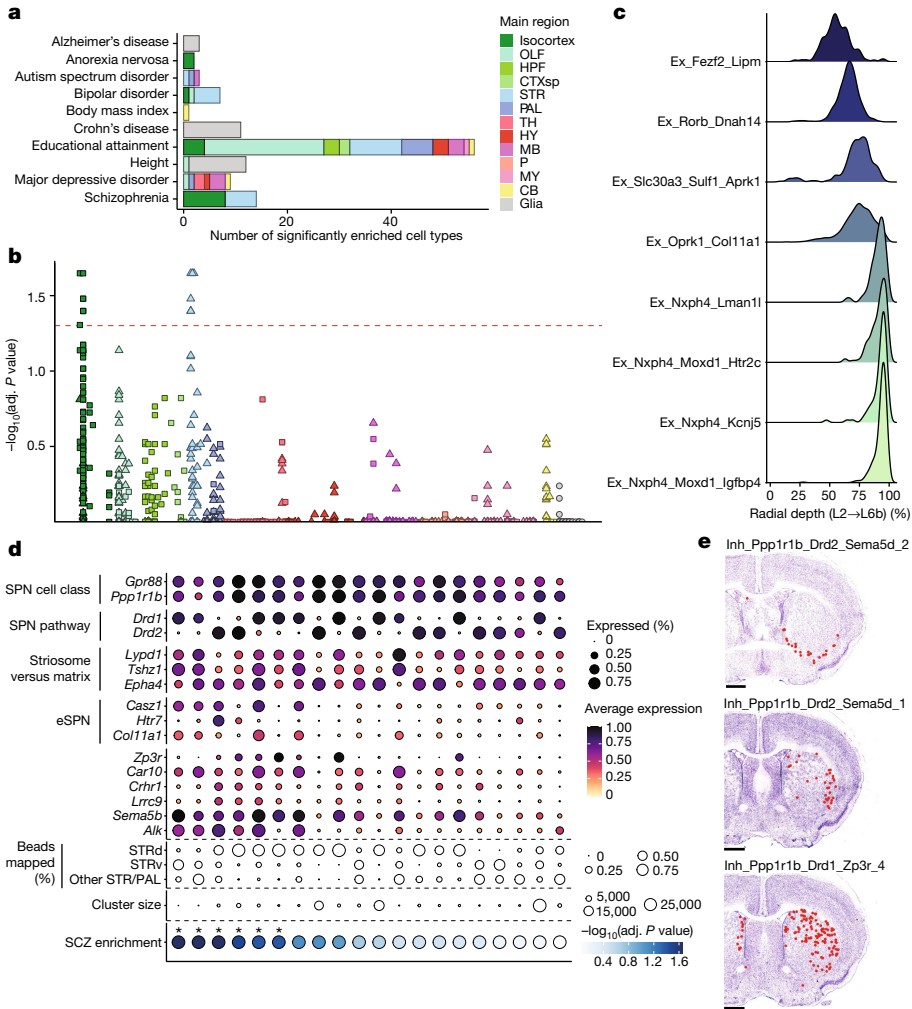

**Fig. 5 | Heritability enrichment for traits studied by GWAS across brain cell types. a,** Bar plots quantifying significantly enriched ($P < 0.05$, computed by single-cell disease relevance score (scDRS) using the one-sided Monte Carlo test, FDR corrected) (Methods) cell types for each trait in non-neurons (grey) and neurons (coloured by main region). **b,** FDR-adjusted (adj.) $-\log_{10} P$ value enrichment scores for each cell type, grouped and coloured by their main regions, for schizophrenia. Squares and triangles denote excitatory and inhibitory clusters, respectively; glia are shown in grey on the far right. $P$ values are computed by scDRS using a one-sided Monte Carlo test. **c,** Ridge plots showing the layer distribution of each excitatory cortical cell type found to be significantly enriched ($P < 0.05$, computed by scDRS using the one-sided Monte Carlo test, FDR corrected) for schizophrenia heritability. **d,** Dot plot of expression of markers of striatal SPN subtype identity grouped by category

(overall cell class identity, pathway identity, matrix versus striosome and eSPN identity). Six additional genes that are enriched in the schizophrenia-enriched ($P < 0.05$, computed by scDRS using the one-sided Monte Carlo test, FDR corrected) SPN types are also shown. STRd, striatum dorsal region; STRv, striatum ventral region; SCZ, schizophrenia. **e,** Representative sections showing the confident mappings of three SPN cell types (confidence value > 0.3) (Methods) significantly enriched ($P < 0.05$, computed by scDRS using the one-sided Monte Carlo test, FDR corrected) for schizophrenia heritability exemplifying Inh_Ppp1r1b_Drd2_Sema5d_2 (top panel; 9 arrays, 178 confident beads total), Inh_Ppp1r1b_Drd2_Sema5d_1 (middle panel; 6 arrays, 119 confident beads total), and Inh_Ppp1r1b_Drd1_Zp3r_4 (bottom panel; 19 arrays, 1,008 confident beads total). Scale bars, 1 mm.

The significance results were robust to using either pseudocells—aggregated collections of cellular neighbourhoods that reduce both computational complexity and noise from statistical dropout (Methods)—or individual cells (Extended Data Fig. 7a). For Alzheimer's disease, heritability enrichment was significant in macrophages and microglia, consistent with analyses of multiple prior datasets[36,47,51]. In autism spectrum disorder, two neuronal cell types showed statistically significant enrichment distributed within the bed nucleus of the stria terminalis, an area with well-established roles in mediating social interactions, and the inferior colliculus, a midbrain structure involved in modulating auditory inputs, a common symptom of patients with autism spectrum disorder. Educational attainment and major depressive disorder—two traits with known high polygenicity—showed enrichment across several regions (Extended Data Fig. 7b).

In schizophrenia (Fig. 5b) and bipolar disorder (Extended Data Fig. 7b), we observed enrichment signals within the excitatory neurons of the isocortex and the inhibitory neurons of the striatum, consistent both with the known shared heritability between these two disorders[52,53] and with prior enrichment studies performed on more limited collections of single-cell datasets[48]. Importantly, although these two signals rose above our stringent threshold for multiple hypothesis testing correction, numerous other subthreshold signals were present, suggesting that these cell type groups are not the only neuronal populations harbouring enrichment for GWAS-associated genes. The significantly enriched excitatory populations were restricted to the lower layers (layers 5 and 6) of cortex (Fig. 5c) and expressed markers suggestive of intratelencephalic and layer 6b identities (Extended Data Fig. 7c). The enriched striatal neuron types all expressed the marker gene *Ppp1r1b*,

identifying them as medium spiny neurons, the principal projection neurons of the dorsal and ventral striatum, which also populate several other pallidal structures. The SPNs can be subdivided by their projection pathway (indirect versus direct), their spatial localization[54] (to the striatal matrix or striosome compartments) or more recently, molecular differences with as yet unclear functional implications[17,55] (called 'eccentric' SPNs versus canonical SPNs). We found that the SPN clusters with the strongest enrichment for schizophrenia heritability expressed markers of an eSPN identity, such as *Casz1*, *Htr7* and *Col11a1*, and were found within both the dorsal and ventral striatum as well as other striatal and pallidal structures (Fig. 5d,e). Together, these results lend additional support to the potential importance of corticostriatal circuitry in the pathogenesis of schizophrenia and highlight the value of a brain-wide atlas for nominating disease-relevant cell types.

## Discussion

Here, we combined snRNA-seq and high-resolution spatial transcriptomics with Slide-seq to generate a comprehensive inventory of cell types across each region of the mouse brain. In total, we identified 4,998 clusters of cells, mostly neuronal, with the diversity distributed primarily in subcortical areas, most especially in the midbrain, pons, medulla and hypothalamus. We utilized the data to uncover specific NP signalling interactions, leveraging the specificity of several NPs and/or their receptors. We also characterized activity-related gene expression patterns across all cell types, identifying conserved genes associated with activity as well as activity-related genes that are more specific to subtypes of neurons. Finally, we nominated specific cell types that are preferentially enriched for the expression of genes associated with human neurological and psychiatric diseases.

We found that interneurons share molecular features with each other across a far wider diversity of neuroanatomical structures than projection neurons, which tend to be more unique to each region. In the cortex and hippocampus, where the functions of interneurons have been studied in the greatest detail, distinct interneuron types are known to have specific circuit roles, such as modulating burst firing, tuning spike timing and mediating disinhibition[56]. Many of these same circuit features are widespread throughout brain areas; for example, local disinhibition modulates respiratory microcircuitry in the medulla[57,58] and fear learning in the amygdala[59]. Interneuron populations may, therefore, maintain more similar molecular identities to serve these common circuit roles, even while the principal projection neurons of individual structures become more specialized. We restricted our analysis to three structures for which the interneuron and projection neuron identities of transcriptionally defined cell types are well known (cortex, striatum and cerebellum); as circuit mapping technologies mature and provide this information for other regions, it will be important to extend these analyses to those areas as well.

A comprehensive inventory of mouse brain cell types should find numerous other immediate uses. One major implication of our analyses is that a substantial fraction of cell types we define are largely unstudied by modern neuroscience methods. To facilitate their interrogation, we deployed an algorithm to identify the minimal set of genes able to specifically define each of our 4,998 clusters. We hope that these genes provide a clear path toward the development of genetic tools that can access a wider portion of the astonishing diversity of the nervous system. Interestingly, we noted a large enrichment of transcription factors amongst the list of genes that most concisely define individual cell types. Combinatorial transcription factor expression is a recurring theme, across central nervous system structures, in the neurodevelopmental specification of diverse neural cell types[60]. Although it is clear in our data that many of these transcription factor combinations represent fixed cell type specifications (based upon our knowledge of how certain transcription factors control development in particular brain areas), additional single-cell data—acquired at different times

of day and in response to different environmental challenges—will be needed to understand which of these clusters represent populations fixed in development and which are more mutable in response to challenges experienced in adulthood.

Beyond achieving more comprehensive access to brain cell types, we anticipate that our dataset will drive computational innovations that better neuroanatomically partition the nervous system and that can integrate other important features of cell type identity, such as connectivity, morphology and physiology. Finally, we expect that our atlas will provide a useful scaffold for interpreting and contextualizing the cell types that are discovered by similar efforts to construct cellular inventories of the human brain[61]. To facilitate these kinds of applications across neuroscience, we have built a portal to visualize, interact with and download these data (www.BrainCell-Data.org). Functions have been implemented to plot gene expression and co-expression in CCF-registered space and within each cell type and to identify genes and cell types enriched within particular brain regions. We also enable the visualization of spatial localizations of each cell type to specific neuroanatomical structures and provide a list of minimum marker genes needed to uniquely distinguish them. We hope that facile access to and interaction with these rich datasets will provide a firm foundation for functionally characterizing the extraordinarily diverse set of cell types that compose the mammalian brain.

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

## Methods

### Animal housing

Animals were group housed with a 12-h light–dark schedule and allowed to acclimate to their housing environment (20–22.2 °C, 30–50% humidity) for 2 weeks post-arrival. All procedures involving animals at Massachusetts Institute of Technology were conducted in accordance with the US National Institutes of Health Guide for the Care and Use of Laboratory Animals under protocol number 1115-111-18 and approved by the Massachusetts Institute of Technology Committee on Animal Care. All procedures involving animals at the Broad Institute were conducted in accordance with the US National Institutes of Health Guide for the Care and Use of Laboratory Animals under protocol number 0120-09-16.

### Brain preparation

At 56 days of age, C57BL/6J mice were anaesthetized by administration of isoflurane in a gas chamber flowing 3% isoflurane for 1 min. Anaesthesia was confirmed by checking for a negative tail pinch response. Animals were moved to a dissection tray, and anaesthesia was prolonged with a nose cone flowing 3% isoflurane for the duration of the procedure. Transcardial perfusions were performed with ice-cold pH 7.4 HEPES buffer containing 110 mM NaCl, 10 mM HEPES, 25 mM glucose, 75 mM sucrose, 7.5 mM MgCl$_2$ and 2.5 mM KCl to remove blood from brain and other organs sampled. For use in regional tissue dissections, the brain was removed immediately; the meninges was peeled away from the entire brain surface, then frozen for 3 min in liquid nitrogen vapour and moved to −80 °C for long-term storage. For use in generation of the Slide-seq dataset through serial sectioning, the brains were removed immediately, blotted free of residual liquid, rinsed twice with OCT to assure good surface adhesion and then oriented carefully in plastic freezing cassettes filled with OCT. These cassettes were vibrated in a Branson sonic bath for 5 min at room temperature to remove air bubbles and adhere OCT well to the brain surface. The brain's precise orientation in the $x$–$y$–$z$ axes was then reset just before freezing over a bath of liquid nitrogen vapour. Frozen blocks were stored at −80 °C.

### Construction of the brain-wide snRNA-seq dataset. Regional dissections.
Frozen mouse brains were securely mounted by the cerebellum or by the olfactory/frontal cortex region onto cryostat chucks with OCT embedding compound such that the entire anterior or posterior half (depending on dissection targets) was left exposed and thermally unperturbed. Dissection of anterior–posterior spans of the desired anatomical volumes was performed by hand in the cryostat using an ophthalmic microscalpel (Feather Safety Razor #P-715) precooled to −20 °C and donning 4× surgical loupes. To microanatomically assess dissection accuracy, 10 µm coronal sections were taken at relevant anterior–posterior dissection junctions and imaged following Nissl staining. Each excised tissue dissectate was placed into a precooled 0.25 ml polymerase chain reaction tube using precooled forceps and stored at −80 °C. Nuclei were extracted from these frozen tissue dissectates within 2 days using gentle detergent-based dissociation as described below.

**Generation of nuclei suspension and construction of snRNA-seq libraries.** Nuclei were isolated from regionally dissected mouse brain samples as previously described[9,62]. All steps were performed on ice or cold blocks, and all tubes, tips and plates were precooled for longer than 20 min before starting isolation. Dissected frozen tissue in the cryostat was placed in a single well of a 12-well plate, and 2 ml of extraction buffer was added to each well. Mechanical dissociation was performed by trituration using a P1000 pipette, pipetting 1 ml of solution slowly up and down with a 1 ml Rainin tip (number 30389212), without creation of froth or bubbles, a total of 20 times. The tissue was allowed to rest in the buffer for 2 min, and trituration was repeated. In total, four or five rounds of trituration and rest were performed. The entire volume of the well was then passed twice through a 26 gauge needle into the same well. Approximately 2 ml of tissue solution was transferred into a 50 ml Falcon tube and filled with wash buffer for a total of 30 ml of tissue solution, which was then split across two 50 ml Falcon tubes (approximately 15 ml of solution in each tube). The tubes were then spun in a swinging-bucket centrifuge for 10 min at 600$g$ and 4 °C. Following spinning, the majority of supernatant was discarded (approximately 500 µl remaining with the pellet). Tissue solutions from two Falcon tubes were then pooled into a single tube of approximately 1,000 µl of concentrated nuclear tissue solution. DAPI was then added to the solution at the manufacturer's (Thermo Fisher Scientific, number 62248) recommended concentration (1:1,000). Following sorting, nuclei concentration was counted using a hemocytometer before loading into a 10X Genomics 3′ V3 Chip.

**snRNA-seq library preparation and sequencing.** The 10X Genomics (v.3) kit was used for all single-nucleus experiments according to the manufacturer's protocol recommendations. Library preparation was performed according to the manufacturer's recommendation. Libraries were pooled and sequenced on NovaSeq S2.

**snRNA-seq reads pre-processing.** Sequencing reads were demultiplexed and aligned to a GRCm39.103 reference using CellRanger v.5.0.1 using default settings (except for an additional parameter to include introns). We used CellBender v.3-alpha[63] to remove cells contaminated with ambient RNA.

### Construction of the brain-wide Slide-seq dataset. Generation of larger surface area Slide-seq arrays.
Slide-seq arrays were generated as previously described[2] with slight modifications. Larger-diameter gaskets were used to generate 5.5 × 5.5 mm$^2$, 6.0 × 6.2 mm$^2$ and 6.5 × 7.5 mm$^2$ bead arrays. These sizes were chosen to facilitate different anterior to posterior coronal section sizes. To facilitate image processing, we utilized 2 × 2 digital binning on the collected data, resulting in 1.3 µm per pixel.

**Serial sectioning procedure.** An OCT embedded P56 wild-type female mouse brain was thermally equilibrated in the cryostat at −20 °C for 30 min and then mounted precisely such that an accurate anatomical alignment was maintained. Just anterior to the end of the olfactory bulb region, a 10-µm-thick coronal slice was set as a starting slide. This starting slide was marked, and the following adjacent 10 µm section was used for Slide-seq library preparation. For each tissue slice used for Slide-seq, a 10 µm pre-slide and a 10 µm post-slide were collected for histology. These histology slides were Nissl stained according to our previously released protocol[64]. After each 10 µm post-slice, an 80 µm gap was trimmed before the next set of serial sections was collected, making each Slide-seq slide interval 100 µm apart. A total of 114 sets of three consecutive slides were collected. All pre- and post-slides for histology registration were stored at −80 °C until the slides were Nissl stained. Optimizations were performed to be able to hold the Slide-seq tissue slices frozen onto their respective pucks at −80 °C during the 2 days required to complete serial sectioning.

**Library generation and sequencing.** Following the serial sectioning procedure, to process multiple samples at the same time, 10-µm-thick tissue slice sections were melted onto Slide-seq arrays and stored at −80 °C for 2 days. On the third day, the frozen tissue sections on the puck were thawed and transferred to a 1.5 ml tube containing hybridization buffer (6× sodium chloride sodium citrate with 2 U µl⁻¹ Lucigen NxGen RNAse inhibitor) for 30 min at room temperature. To generate libraries, the Slide-seqV2 protocol was adapted from the previously published Slide-seqV2 protocol[2,65], in which the volume of reagents was scaled to accommodate the larger surface array of the arrays. Libraries were sequenced using the standard Illumina protocol. The samples were sequenced on either NovaSeq 6000 S2 or S4 flow cells at a depth of 1.1–1.5 billion reads per array, adjusting for the array size. Samples were pooled at a concentration of 4 nM and followed the read structure previously described[2].

**Imaging of Nissl sections.** We acquired Nissl images on an Olympus VS120 microscope using a ×20, 0.75 numerical aperture objective. Images were captured with a Pike 505C VC50 camera under autoexposure mode with a halogen lamp at 92% power. The pixel size in all images was 0.3428 μm in both the height and width directions. We acquired a total of 114 Nissl images, each from an adjacent section of the brain to a corresponding section that was processed using the Slide-seq pipeline. Of the 114 sections, we removed 10 from the posterior medulla and upper spinal cord that were outside of the area of the CCF reference brain. Of the remaining 104 images, we removed an additional three sections because of the unsatisfactory quality of the corresponding Slide-seq puck data. The remaining 101 images comprise the final dataset that we use for all our analyses.

**Slide-seq reads pre-processing.** The sequenced reads were aligned to GRCm39.103 reference and processed using the Slide-seq tools pipeline (https://github.com/MacoskoLab/slideseq-tools; v.0.2) to generate the gene count matrix and match the bead barcode between array and sequenced reads.

**Registration of Slide-seq data to CCF. Alignment of Slide-seq arrays to adjacent Nissl sections.** As a pre-processing step for the alignment of Slide-seq arrays to Nissl images, for each puck we generated a greyscale intensity image from the Slide-seq data by summing the UMI counts (across all genes) at each bead location on the puck and normalizing by the maximum UMI count value across the entire puck. We then performed the alignment of these images to the adjacent Nissl images in two steps. First, we transformed each Nissl image to an intermediate coordinate space using a manual rigid transformation. The purpose of this first transformation is to bring all the Nissl images to an approximately equivalent upright orientation, which made the second step of alignment easier. In the second step, we manually identified corresponding fiducial markers in the Nissl images and Slide-seq intensity images using the Slicer3D tool v.4.11 (ref. 66) along with the IGT fiducial registration extension[67]. We then computed the bead positions for all beads through thin-plate spline interpolation, where the spline parameters were determined using the fiducial markers.

**Alignment of Nissl sections to the CCF.** Our series of Nissl sections, downsampled to 50 μm resolution by local averaging, were aligned to the 50 μm CCF by jointly estimating three transformations. First, a three-dimensional diffeomorphism modelled any shape differences between our sample and the atlas brain. This transformation is modelled in the Large Deformation Diffeomorphic Metric Mapping framework[68]. Second, a three-dimensional affine transformation (12 degrees of freedom) modelled any pose or scale differences between our sample and the deformed atlas. Third, a two-dimensional rigid transformation (three degrees of freedom per slice) on each slice modelled positioning of samples onto microscopy slides.

Dissimilarity between the transformed atlas and our imaging data was quantified using an objective function we developed previously[69,70], equal to the weighted sum of square error between the transformed atlas and our dataset, after transforming the contrast of the atlas to match the colour of our Nissl data at each slice. To transform contrasts, a third-order polynomial was estimated on each slice of the transformed atlas to best match the red, green and blue channels of our Nissl dataset (12 degrees of freedom per slice). During this process, outlier pixels (artifacts or missing tissue) are estimated using an expectation maximization algorithm, and the posterior probabilities that pixels are not outliers are used as weights in our weighted sum of square error.

This dissimilarity function, subject to Large Deformation Diffeomorphic Metric Mapping regularization, is minimized jointly over all parameters using a gradient-based approach, with estimation of parameters for linear transforms accelerated using Reimannian gradient descent as recently described[71]. Gradients were estimated automatically using pytorch, and source code for our standard registration pipelines is available online at https://github.com/twardlab/emlddmm. The transformations above were used to map annotations from the CCF onto each slice. The boundaries of each anatomical region were rendered as black curves and overlaid on the imaging data for quality control. We visually inspected the alignment accuracy on each slice and identified 15 outliers, where our rigid motion model was insufficient owing to large distortions of tissue slices. For these slices, we included an additional two-dimensional diffeomorphism to model distortions that are independent from slice to slice and cannot be represented as a three-dimensional shape change, as in our previous work[72]. Extended Data Fig. 2a shows accuracy before and after applying the additional two-dimensional diffeomorphism.

**CCF groups used in visualization.** For ease of visualization, we grouped the CCF hierarchy into 12 'main regions': isocortex, olfactory areas (OLF), hippocampal formation (HPF), striatum (STR), pallidum (PAL), hypothalamus (HY), thalamus (TH), midbrain (MB), pons (P), medulla (MY) and cerebellum (CB). For many of our analyses, we also grouped into 'DeepCCF' regions, detailed in Supplementary Table 4.

**Analysis of CCF accuracy.** We analysed three genes with highly stereotyped and regional expression, *Dsp*, *Ccn2* and *Tmem212,* which correspond to the CCF regions detailed in Supplementary Table 12.

For each bead with non-zero expression of the specified genes, we calculated the distance to the corresponding CCF regions. For preliminary quality control, we used the dbscan package[73] with eps=3 to filter the points and used the full width at half maximum metric to summarize the distances (Extended Data Fig. 2c).

**Clustering of snRNA-seq data. Overview.** Clustering was performed hierarchically starting from the full dataset of approximately 6 million single nuclei. Each round of clustering consisted of (1) gene selection based on a binomial model; (2) square-root transformation of the counts; (3) construction of the $k$ nearest neighbour and shared neighbour graphs; and (4) Leiden clustering over a range of resolution parameters to find the lowest resolution that yielded multiple clusters. The resulting clusters were then each iteratively re-clustered, and the process was repeated until either (1) no Leiden resolution resulted in a valid clustering or (2) the resulting clusters did not have at least three differentially expressed genes distinguishing them. A key goal of this clustering strategy was to re-calculate gene selection for every clustering, as the relevant variable genes depend on the overall context of the cells being clustered. This resulted in a distributed design in which the data were stored on a disk in a compressed representation that could be efficiently accessed using parallel processes. This allowed us to perform clustering thousands of times without creating redundant copies of the data.

**Variable gene selection.** To identify variable genes, we used a binomial model of homogenous expression and looked for deviations from that expectation, similar to a recently described approach[74]. Specifically, for each gene we computed the relative bulk expression by summing the counts across cells and dividing by the total UMIs of the population. This is the proportion of all counts that are assigned to that gene. We use this value as $p$ in a binomial model for observing the gene in a cell with $n$ counts (equivalently, $np$ is equivalent to $\lambda$ in a Poisson model). The expected proportion with non-zero counts is thus

$$P(x > 0) = 1 - e^{-\lambda}.$$

We compared this expected value with the observed percentage of non-zero counts and selected all genes that are observed at least 5% less than expected in a given population.

**Construction of shared nearest neighbour graphs.** After selecting variable genes, we constructed a shared nearest neighbour graph[75,76]. First, we transformed the counts with the square-root function and then computed the $k$-nearest neighbour ($k$NN) graph using cosine distance and $k = 50$ (not including self-edges). From the $k$NN graph, we

compute the shared neighbour graph, where the weight between a pair of cells is the Jaccard similarity over their neighbours:

$$J(A, B) = \frac{|A \cap B|}{|A \cup B|},$$

where $A$ and $B$ represent the sets of neighbours for two cells in the $k$NN graph.

**Leiden clustering.** Once we computed the shared nearest neighbour graph, we used the Leiden algorithm to identify cell clusters using the Constant Potts Model for modularity[77]. This method is sensitive to a resolution parameter, which can be interpreted as a density threshold that separates intercluster and intracluster connections. To find a relevant resolution parameter automatically, we implemented a sweep strategy. We started with a very low-resolution value, which results in all cells in one cluster. We gradually increased the resolution until there were at least two clusters and the size ratio between the largest and second-largest cluster was at most 20, meaning that at least 5% of the cells are not in the largest cluster. Any cluster of fewer than $\sqrt{N}$ cells was discarded, where $N$ was the number being clustered in that round. This discarded set constituted roughly 1.6% of the total cells (100,280 of 5.9 million).

**Clustering termination and marker gene search.** The clustering strategy described above was applied recursively on the leaves of the tree until one of the following conditions was met.

- If the shared neighbour graph was not a single connected component, there is no resolution low enough to form a single cluster, and so, the resolution sweep was not possible. This would typically occur if there were very few variable genes, which is indicative of a homogenous cell population.
- If the resolution sweep concluded at the highest resolution without ever finding multiple clusters, this is also indicative of a homogenous population, and clustering was considered completed.
- Finally, we truncated the tree when the resulting clusters did not have differentially expressed markers that defined them.

To test for differential markers, we considered each leaf versus its sibling leaves. We used a Mann–Whitney $U$-test to assess whether any genes are differentially expressed. As an additional filter, we required that a gene be observed in less than 10% of the lower population and observed at a rate at least 20% higher in the higher population to ensure that there is a discrete difference in expression between the two populations. We required every cluster to have at least three marker genes distinguishing it from its neighbours as well as three marker genes in the other direction. If a cluster failed that test, all leaves were merged, and the parent was considered the terminal cluster.

The only exception to the above was if the next level of clustering resulted in a set of differential clusters that passed this test; these were situations where the first round of clustering split the cells on a continuous difference in expression but the next round resolved the discrete clusters. We retained these clusters for further subclustering as they may contain additional structure.

**Visualization of clusters.** For high-dimensional visualization, as in Fig. 1a, we first subsampled each of the clusters to a maximum of 2,000 nuclei. Using the Scanpy package, we calculated the first 250 principal components of our subsampled cells. We then ran OpenTSNE v.1.0.0 (ref. 78) on the principal component space to generate a $t$-SNE that optimizes both local and global structure using an exaggeration factor of four and a perplexity of 350.

**Visualization of cluster gene expression.** For the heat map visualization in Fig. 1c, we subsetted the 1,937 mapped cell types to the 1,260 neuronal cell types with at least five confidently mapped beads in at least one puck. We normalized the data with Seurat's LogNormalize normalization (scale.factor=1e4) and averaged each cell type's five nearest neighbours' expressions. The main region assignment was

determined by combining the 10 nearest neighbours' imputed main region assignment. The matrix was plotted using the ComplexHeatmap package in R[79].

**Quality control of clusters.** A strict, multistep quality assessment framework was used to retain only high-quality cell profiles in our analyses. First, we removed nuclei with less than 500 UMIs and greater than 1% mitochondrial UMIs. Doublet clusters were further flagged and excluded based on co-expression of marker genes of distinct cell classes (Supplementary Table 2) (for example, *Mbp* and *Slc17a7*).

Next, we constructed a cell 'quality network' to systematically identify and remove remaining low-quality cells and artefacts from the dataset. By simultaneously considering multiple quality metrics, our network-based approach has increased power to identify low-quality cells while circumventing the issues related to setting hard thresholds on multiple quality metrics. To construct the quality network, we considered the following cell-level metrics: (1) per cent expression of genes involved in oxidative phosphorylation; (2) per cent expression of mitochondrial genes; (3) per cent expression of genes encoding ribosomal proteins; (4) per cent expression of IEG expression; (5) per cent expression explained by the 50 highest expressing genes; (6) per cent expression of long non-coding RNAs; (7) number of unique genes log$_2$ transformed); and (8) number of unique UMIs (log$_2$ transformed). Given their inherently distinct distributions of quality metrics, we separately constructed quality networks for neurons and glial cells. The quality network was constructed and clustered using shared nearest neighbour and Leiden clustering (resolution 0.8) algorithms from Seurat v.4.2.0. Our strategy was to remove any cluster from the quality network with 'outlier' distribution of quality metric profiles. A distribution of quality metric was considered as an outlier if its median was above 85% of cells in three features of the quality network: oxidative phosphorylation, mitochondrial and ribosomal protein expression. We further removed any remaining clusters with fewer than 15 cells.

**Estimation of snRNA-seq sampling depth.** We used the R package SCOPIT v.1.1.4 (ref. 80) to estimate the sequencing saturation of our dataset. Under the prospective sequencing model, SCOPIT calculates the multinomial probability of sequencing enough cells, $n^*$, above some success probability, $p^*$, in a population containing $k$ rare cell types of size $N$ cells, from which we want to sample at least $c$ cells in each cell type:

$$n^* = \min\{n \mid P(N_1 \geq c, N_2 \geq c, ..., N_k \geq c) \geq p^*\}.$$

We assume there are $k = 19$ rare cell types in our population of mapped cells, each containing $N = 101$ cells (frequency of 0.0024% amongst all mapped cell types). We need to sequence at least $c = 81$ cells from each cell type for sufficient sampling (80% of the rarest cell type). We used SCOPIT to estimate the sampling saturation of our mapped dataset of 4,210,212 cells, and then, we used the same sampling curve to estimate saturation of our full dataset (mapped and unmapped) of 4,388,420 cells.

**Note about immune cell types.** We identified 16 cell classes in our snRNA-seq data, 6 of which were excluded from the majority of our analyses (dendritic cell, granulocyte, lymphocyte, myeloid, olfactory ensheathing and pituitary). Most of these excluded clusters are classified as immune cell types and are mentioned in the following figure and tables: Extended Data Fig. 1a,d,h and Supplementary Tables 2 and 3. In addition, we mapped many immune cell populations.

**Cell type mapping into the Slide-seq dataset with RCTD.** We used RCTD to map the single-nuclei clusters onto the Slide-seq spatial beads.

For mapping we deployed a modification of the RCTD algorithm[23], in which we increased the computational efficiency and throughput, modified cell type prefiltering and adjusted the metric used for the decomposition assignment (see below).

**Changes to RCTD for parallelizable throughput.** We changed the quadratic programming optimizer of RCTD to use OSQP[81], which scales better for the larger matrices resulting from larger sets of cell types to be mapped. We also rewrote the inner loops of the most time-intensive functions (choose_sigma_c and fitPixels) with Rcpp[82] for efficiency. Additionally, we used Hail Batch (refs. 83,84) and GNU Parallel[85], which allowed for large-scale, on-demand parallelization (to thousands of cores) using cloud computing services.

**Changes to RCTD for cell type prefiltering.** RCTD in doublet mode models how well explicit pairs of cell types match a bead's expression. For computational efficiency, RCTD prefilters which cell type pairs are considered per bead. However, we found that larger cell type references with many similar cell types led to overly sparse prefiltering, which impeded our ability to confidently map fine-grained cell types. To balance this sparsity, we added an additional ridge regression term to RCTD's quadratic optimization tunable with a ridge strength parameter, which allowed us to control the relative sparsity and potential overfitting of the prefiltering stage. Our modified prefiltering stage used a heuristic to detect a subset of potential cell types for each bead by using RCTD's full mode with two ridge strength parameters (0.01, 0.001), as well as mapping each cell type individually.

In accordance with the explicit cell type pairs used within RCTD's doublet mode, we subdivided this filtered list, pulling out the 10 cell types deemed most likely to be associated with the given bead. When modelling how well these cell types mapped to a given bead, we exhaustively used one cell type from the top 10 list and one cell type from the rest of the prefiltered list. For the cerebellum and striatum, the number of cell types considered was sufficiently low that we were able to run the algorithm using all pairs.

**Changes to RCTD for decomposition assignment.** To aid in mapping large references with many similar clusters, we modified how RCTD scores explicit pairs of cell types in doublet mode. Rather than using the result of the single-cell type pair that fit best, we identified the cell type pairs that scored similar to the best-scoring pair (with likelihood score within 30). Then, we collated the frequency of each cell type occurring in these well-fitting pairs and divided by the total occurrences of all the cell types to make a confidence score. Throughout the paper, we use 0.3 (of a maximum score of 0.5) as the threshold for a 'confident' mapping.

**Creation of per-region cell type references and gene lists.** To help reduce the computational load of combinatorially mapping the cell types to each bead, we created a set of tailored references for each region. First, we grouped the libraries into at least one of eight large-scale regions corresponding to (1) the basal ganglia; (2) medulla and pons; (3) cerebellum; (4) hippocampal formation; (5) isocortex; (6) midbrain; (7) olfactory bulb; and (8) striatum. For each reference region, the clusters used for mapping had a minimum of 50 cells from the aforementioned per-region libraries and at least 100 cells total.

For each reference region, we also generated a tailored gene list. First, for each cluster in each reference region, we ran the same Mann–Whitney $U$-test as in the cluster generation (see above), where the background expression was the other clusters in the reference set. Then, we combined all results per gene and chose the 5,000 genes with the smallest $P$ value across all the individual differential expression tests.

**Running RCTD on per-region puck subsets.** We assigned the CCF regions into at least one of the eight large-scale regions from above. Then, for each Slide-seq puck, we grouped the beads on the puck into at least one of the large-scale regions using our CCF alignment. For each large-scale region on each puck, we ran RCTD using the corresponding tailored reference cell types and tailored gene list. We additionally considered only beads that had at least 150 UMIs across all genes and at least 20 UMIs within the tailored gene list.

**Constructing and analysing cell type dendrogram. Constructing Paris dendrogram and aggregation into groups.** To build a graph of cell type similarity, we used Scanpy on our subsampled data to compute the connectivities over a 20 neighbour local neighbourhood using 250 principal components (the section 'Visualization of clusters' has details about subsampling). We aggregated this weighted adjacency matrix row and column wise by taking the average weights of all cells in a given cell type. We then used the Paris hierarchical clustering algorithm from scikit-network v.0.28.1 to build a dendrogram from our cell type adjacency matrix[86]. We plotted major cell type markers and examined spatial localization patterns to organize our neuronal clusters into larger sets, comprising a total of 223 groups (metaclusters). Using Scanpy's rank_genes_groups with the Wilcoxon method, we generated a table of the top 50 differentially expressed genes per metacluster (Supplementary Table 6).

**Reordering dendrogram.** Given this tree structure, we optimized the leaf node sequence in the tree by selectively swapping the order of the children of internal nodes. We did so by iteratively permuting the columns and rows of a normalized cell type by gene matrix so that the elements are grouped around the diagonal. The genes *Tbr1*, *Fezf2*, *Dlx1*, *Lhx6*, *Foxg1*, *Neurod6*, *Lhx8*, *Sim1*, *Lmx1a*, *Lhx9*, *Tal1*, *Pax7*, *Hoxc4*, *Gata3*, *Hoxb5* and *Phox2b* were chosen to be discrete, biologically interpretable markers—mostly transcription factors that relate to overall neuronal cell lineage.

The genes and cell types were initially reordered using the R package slanter's default permuting method[87]. The cell types were then reordered to comply with the cell type dendrogram structure using a dynamic programming tree-crossing minimization optimization[88].

**Finding proximate neighbourhoods within dendrogram.** Given an index neuronal cell type, to find its proximate neighbourhood within the dendrogram, we consecutively aggregated descendants from successively more distant ancestors. We continued aggregating until the number of cell types in the neighbourhood would surpass 100 or for neurons, if the next set of cell types was more than 60% non-neuronal.

**Analyses of cluster heterogeneity across regions. Cell types needed for 95% beads.** To assess cluster heterogeneity across regions with vastly different areas, we analysed the minimum number of cell types required to cover 95% of the mapped beads. For each region, we computed the number of confidently mapped beads for each cell type sorted in descending order by the number of beads. Next, we determined the number of cell types necessary for the running sum of beads to reach 95% of the total mapped beads.

**Force-directed DeepCCF region graph.** To generate the force-directed graph of regional cell type similarity, as in Fig. 2b, we weighted each pair of DeepCCF regions with the weighted Jaccard similarity metric. We then used the R package qgraph v.1.9 to generate a force-directed graph.

**Projection and interneuron ridge plots.** To generate the neighbourhood ridge plots in Fig. 2c, we first identified the interneuron and projection metaclusters for the isocortex, striatum and cerebellum, detailed in Supplementary Table 13. Supplementary Table 5 shows the cell types within each metacluster.

**Discovery of combinatorial marker genes needed to distinguish snRNA-seq cell types.** To find the minimally sized gene lists that allowed us to distinguish one cell type from the others in the dataset, we framed the question as a set covering problem. In the set cover problem, we find the smallest subfamily of a family of sets that can still cover all the elements in the universe set. We can define this as a mixed integer linear programming model programmatically using the JuMP domain-specific modelling language in Julia (refs. 33,89). We optimized using the HiGHS open-source solver (v.1.5.1)[90] or the IBM ILOG CPLEX commercial solver v.22.1.0.0 (ref. 91). Supplementary Methods has the mixed integer linear programming model derivation and CPLEX solver parameters used.

**Neurotransmitter and NP assignment to cell types. Neurotransmitter assignment.** Each cell type was assigned to a neurotransmitter

identity based upon the percentage of its cells with non-zero counts of genes essential for the function of that neurotransmitter. Specifically, we used a non-zero threshold $nz = 0.35$.

- VGLUT1: $Slc17a7 \geq nz$
- VGLUT2: $Slc17a6 \geq nz$
- VGLUT3: $Slc17a8 \geq nz$
- GABA: $(Gad1|Gad2 \geq nz)$ and $(Slc32a1 \geq nz)$
- GLY: $(Gad1|Gad2 \geq nz)$ and $(Slc6a5|Slc6a9 \geq nz)$
- CHOL: $Slc18a3$ and $Chat \geq nz$
- DOP: $Slc6a3 \geq nz$
- NOR: $Pnmt|Dbh \geq nz$
- SER: $Slc6a4|Tph2 \geq nz$

For the 166 neuronal cell types that did not meet the above $nz$ conditions, we carefully examined their top expressing transporters and assigned neurotransmitters accordingly.

**NP assignment.** Each cell type was assigned to an NP ligand identity if (1) the percentage of its cells with non-zero expression of the NP was greater than or equal to 0.3 and (2) the average expression of the NP was greater than or equal to 0.5 counts per cell. We observed that the expression of four NPs showed greater contamination across other cell types: OXT, AVP, PMCH and AGRP. Therefore, for these NPs, we required the percentage of cells with non-zero expression to be greater than or equal to 0.8 and average expression to be greater than or equal to five counts per cell.

Each cell type was assigned to an a neuropeptide receptor (NPR) identity if (1) the percentage of its cells with non-zero expression of at least one NPR was greater than or equal to 0.2 and (2) the average expression of at least one NPR was greater than or equal to 0.5 counts per cell.

**Quantification of region-specific excitatory–inhibitory ratios.** We first created inhibitory and excitatory cell type groups based on their neurotransmitter expression as above. We classified cell types expressing GABA or GLY neurotransmitters as inhibitory and those expressing VGLUT neurotransmitters as excitatory. In the case where a cell type was assigned to both an inhibitory identity and an excitatory identity, it was classified as inhibitory. For each region on the Slide-seq array, we labelled its beads as excitatory or inhibitory by whether they confidently mapped into members of the corresponding cell type groups, with additional filtering to ensure that these mappings were one of top two ranked cell types per bead. Then, defining $\#I$ and $\#E$ as the number of inhibitory and excitatory mapped beads, respectively, we defined the excitatory-to-inhibitory fraction as $\frac{\#I}{\#I + \#E}$.

To quantify the uncertainty, we calculated the 95% confidence interval for the corresponding binomial distribution using the exact method of binconf function in the Hmisc R package[92]. For plotting clarity, regions with fewer than five total inhibitory and excitatory cells were excluded.

**Analyses of activity-dependent gene expression. Pseudocell generation.** Using scOnline, we aggregated our snRNA-seq expression data into pseudocells: aggregations of cells with similar gene expression profiles. Working at the pseudocell resolution (rather than with individual cells) eliminates the technical variation issues of single-cell transcriptomic data, such as low capture rate from dropouts and pseudoreplication through averaging expression of similar cells[93,94], while avoiding issues of pseudobulk approaches, such as low statistical power and high variation in sample sizes[95].

To generate our pseudocells, we first performed dimensionality reduction at the single-cell level. Single cells were divided into 27 groups, consisting of glial cell classes and neuronal populations further divided by neurotransmitter usage. Within each cell group, we selected genes that were highly variable in a specific number of mouse donors such that a maximum of 5,000 genes would be used for subsequent scaling by batch. We then ran principal component

analysis on the scaled expression data (50 principal components for glia and 250 principal components for neurons). Next, we constructed pseudocells by grouping single cells within each cell type. Within a cell type of size $n$, cells were assigned to pseudocells of size $s$ such that the pseudocell size correlated with cell type size:

$$s = \min\left(200, \max\left(20, \frac{n}{50}\right)\right).$$

Pseudocell centres were identified by applying $k$-means clustering on the top principal components (50 for glia and 250 for neurons). To ensure the stability of results across different cell type sizes—ranging from rare neuronal clusters of 15 cells to a glial cluster of half a million cells—we weighted principal components by their variance explained. Random walk approaches have been found to be more robust in identifying cells of similar gene expression profiles in contrast to spherical, distance-based methods[96,97]. Therefore, we used the random walk method on cell–cell distances in the principal component analysis space to assign cells to pseudocell centres (that is, $k$-means centroids)[97]. To generate our pseudocell counts matrix, we aggregated the raw UMI counts of cells assigned to each pseudocell. This resulted in representation of each cell type by one or more pseudocells, ranging from 1 to 2,490 pseudocells.

The pseudocell-level expression of protein-coding genes was normalized by $\log_2$-transformed count per million followed by quantile normalization. We further normalized the expression of each gene to have a mean expression of zero and a standard deviation of one. Normalized pseudocell counts were used for downstream analysis.

**Candidate ARG list.** To generate our candidate ARGs list, we divided our neuronal pseudocells into 28 cell groups such that each main region was assigned to an excitatory and inhibitory population, and all other cell types (cholinergic, dopaminergic, noradrenergic and serotonergic) were individually grouped across all main regions. To construct gene co-expression networks within each cell group, we computed pairwise gene correlation coefficients (Pearson) across scaled pseudocells using the R package psych v.2.2.5. For a gene $g$ to be considered an ARG candidate, its correlation $r$ with $Fos$ must be the following: (1) $r$ is greater than or equal to 0.3; (2) $r$ in the greater than or equal to 99.5% quantile distribution of $g$'s correlations with all genes; and (3) $r$ is statistically significant after multiple hypothesis testing (Holm-adjusted $P < 0.05$). To construct the final full ARG candidate list, we took the union of selected genes across all cell groups.

**ARG network.** To identify activity-regulated relationships between our candidate ARGs and regions of the brain, we constructed a force-directed graph of a weighted bipartite network. We used the R package igraph v.1.2.7 to build the network from an incidence matrix of candidate ARGs and excitatory/inhibitory cell types localized to different regions. An entry $e$ in the matrix corresponds to a gene's correlation $r$ with $Fos$ in a brain region scaled up by one such that all entries are greater than or equal to one. The nodes of the network comprised two disjoint sets, candidate ARGs and neuronal brain regions, such that there would never be an edge between a pair of genes or a pair of regions. Edges were weighted based on the correlation entry $e$ between a gene and region node. To emphasize the most central nodes in the network, we pruned edges with $e$ less than 1.3. We then calculated the degree of each node in the pruned network and selected our core IEGs from the network based on node centrality (degree > 18).

**Classifying ARG clusters.** To further characterize our candidate ARGs, we performed ward.D2 hierarchical clustering based on their $Fos$ correlations across brain regions. We cut the dendrogram at a height that divided our ARGs into seven clusters. To assess the overlap between our ARG clusters and the ARGs reported in Tyssowski et al.[42], we computed a Fisher's exact test between two given gene sets using the R package

GeneOverlap v.1.30.0 (ref. 98). *P* values were Bonferroni corrected for multiple hypothesis testing in each gene cluster.

**Transcription factor enrichment.** To identify the transcription factors selectively enriched for telencephalic excitatory or inhibitory populations, we performed gene set enrichment analysis (GSEA) on a ranked gene list against a curated transcription factor gene set. First, we ranked genes from our telencephalic excitatory cells by their average correlation with *Fos* compared with a background ranking of average telencephalic inhibitory *Fos* correlations. Genes from our telencephalic inhibitory cells were ranked in reverse order. We built a gene set of transcription factors by combining enrichR[99] databases from ARCHS4, ENCODE and TRRUST. We subsetted for human transcription factors that showed up in at least two databases and had at least ten unique targets in each of those databases. The final gene set consisted of these human transcription factors with targets composed of the intersection between any two databases. We used the R package fgsea v.1.20.0 (ref. 100) to run gene set enrichment analysis with a gene set size restriction of 15–500 and against a background of all protein-coding genes expressed in our normalized pseudocell data. *P* values were computed using a positive one-tailed test and FDR corrected by fgsea.

**Heritability enrichment with scDRS.** To determine which of our snRNA-seq cell types was enriched for specific GWAS traits, we used scDRS (v.1.0.2)[50] with default settings. scDRS operates at the single-cell level to compute disease association scores, while considering the distribution of control gene scores to identify significantly associated cells.

We used MAGMA v.1.10 (ref. 101) to map single-nucleotide polymorphisms to genes (GRCh37 genome build from the 1000 Genomes Project) using an annotation window of 10 kb. We used the resulting annotations and GWAS summary statistics to calculate each gene's MAGMA *z* score (association with a given trait). Human genes were converted to their mouse orthologs using a homology database from Mouse Genome Informatics (MGI). The 1,000 disease genes used for scDRS were chosen and weighted based on their top MAGMA *z* scores. Many of the traits we tested for enrichment had previously computed MAGMA *z* scores[50], so those scores were used instead (after applying MGI gene ortholog conversion).

scDRS was used to calculate the cell-level disease association scores for a given trait; in our case, we treated our aggregated raw pseudocell counts as the input single-cell dataset, validating that the pseudocell results largely recapitulated single cell-level results for three traits (Extended Data Fig. 7a). To determine trait association at the annotated cell type resolution, we used the *z* scores computed from scDRS's downstream Monte Carlo test. These Monte Carlo *z* scores were converted to theoretical *P* values using a one-sided test under a normal distribution. Theoretical *P* values were FDR corrected for multiple hypothesis testing, considering only cell types with at least four beads confidently mapped to a single puck and deep CCF region, as well as non-neurogenesis cell types.

### Reporting summary

Further information on research design is available in the Nature Portfolio Reporting Summary linked to this article.

### Data availability

Raw single-nucleus RNA sequencing and Slide-seq data are available in the Neuroscience Multi-omic Archive (www.NeMOArchive.org; Research Resource Identification: SCR_016152) under identifier nemo:dat-aa0jwmj. The genomic reference (GRCm39.103), aligned data (single-nucleus RNA sequencing and Slide-seq), integrated data and Nissl stain images are available at www.BrainCellData.org, where additional interactive visualizations are also available.

### Code availability

The single-nucleus RNA sequencing clustering algorithm, Robust Decomposition of Cell Type Mixtures modifications and analysis code are available in the code repository at www.github.com/MacoskoLab/brain-atlas with accompanying package versions detailed.

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

**Acknowledgements** We thank G. Fishell, T. Kamath, W. Regehr and J. Welch for helpful discussions. We also thank J. Goldstein, D. King and the rest of the Hail Batch team for giving computational assistance. This work was supported by the National Institutes of Health/National Institute of Mental Health (Brain Grants 1U19MH114821 to E.Z.M. and RF1MH124598 to F.C. and E.Z.M.) as well as the Stanley Center for Psychiatric Research.

**Author contributions** F.C. and E.Z.M. conceived the study. J.L. and N.S.S. led the analyses with help from V.G and D.M.C. J.T.W. developed the single-nucleus RNA sequencing clustering algorithm. M.R. built the data portal and aligned the adjacent Nissl images to the Slide-seq array data. D.T., C.M. and X.L. performed the Allen Common Coordinate Framework integration under supervision from P.M. N.M.N. and C.V. led the single-nucleus RNA sequencing data generation with help from T.N. K.S.B. led the Slide-seq data generation with help from E.M. and C.V. under supervision of F.C. K.F. assisted with study design and implementation. J.L., N.S.S. and E.Z.M. wrote the paper with contributions from all authors.

**Competing interests** F.C. and E.Z.M. are academic founders of Curio Bioscience.

**Additional information**
**Correspondence and requests for materials** should be addressed to Fei Chen or Evan Z. Macosko.

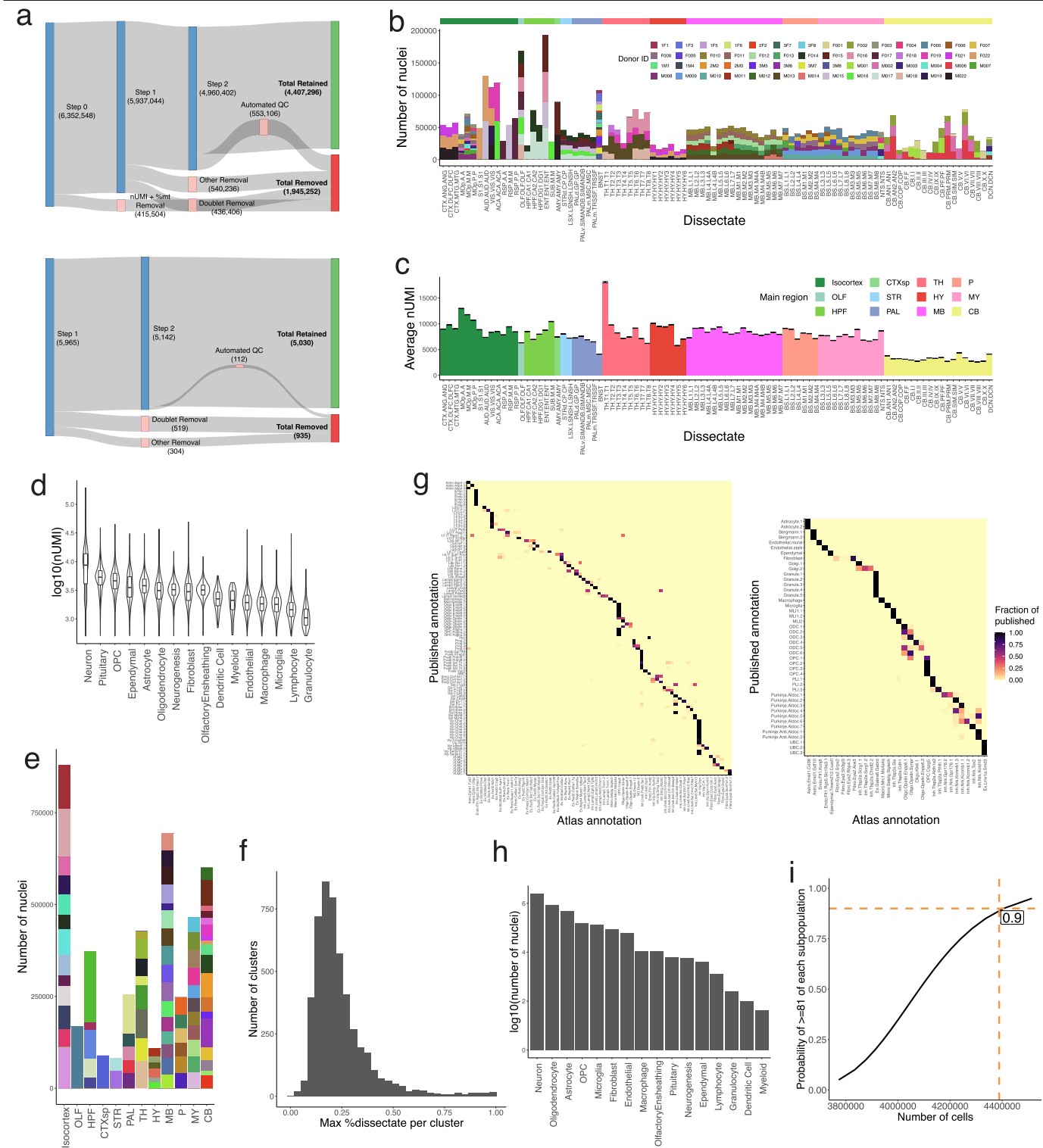

**Extended Data Fig. 1 | Quality control and summary statistics of the snRNA-seq analysis. a**, Sankey diagrams showing the number of nuclei (top) and clusters (bottom) retained and removed at each step of our quality control workflow. The final retained numbers include immune cells. mt (mitochondrial gene). **b**, Stacked bar plots showing the nuclei sampled per region, for each animal replicate. Female donor IDs contain an "F", while male donors contain an "M". The top colouring indicates the dissectate's major region. **c**, Bar plots showing the average UMIs per nucleus in each dissectate, coloured by main region. Error bars indicate standard error of each dissectate's average number of UMIs. n = 4388420 nuclei examined over 92 dissectates. **d**, Violin plots showing the log10 distribution of the UMIs per nucleus in each major cell class (including immune cell classes). n = 4407296 nuclei examined over 16 cell classes. Box plots centred at median, bounded by IQR (25-75th percentile),

with lower whisker at data point >= (25th percentile − 1.5*IQR) and upper whisker at data point <= (75th percentile + 1.5*IQR). **e**, Stacked bar plots of the nuclei sampled in each major mouse brain region, sub-setted by individual dissectate. **f**, Histogram of the maximal proportional representation of individual dissectates in each snRNA-seq cluster. **g**, Heatmap representing a confusion matrix between clustering of the snRNA-seq data in the current study (x-axis), and published studies (y-axis), for the mouse motor cortex[6] (left) and cerebellum[9] (right). **h**, Histogram of the log10 nuclei recovered from each major cell class (including immune cell classes). **i**, Plot indicating the probability of sampling 19 very rare populations (prevalence 0.0024% among all mapped cell types) as a function of the total number of mapped cells profiled in experiment (probability estimation in Methods). Number of high-quality nuclei profiled here (4388420) and corresponding probability are indicated.

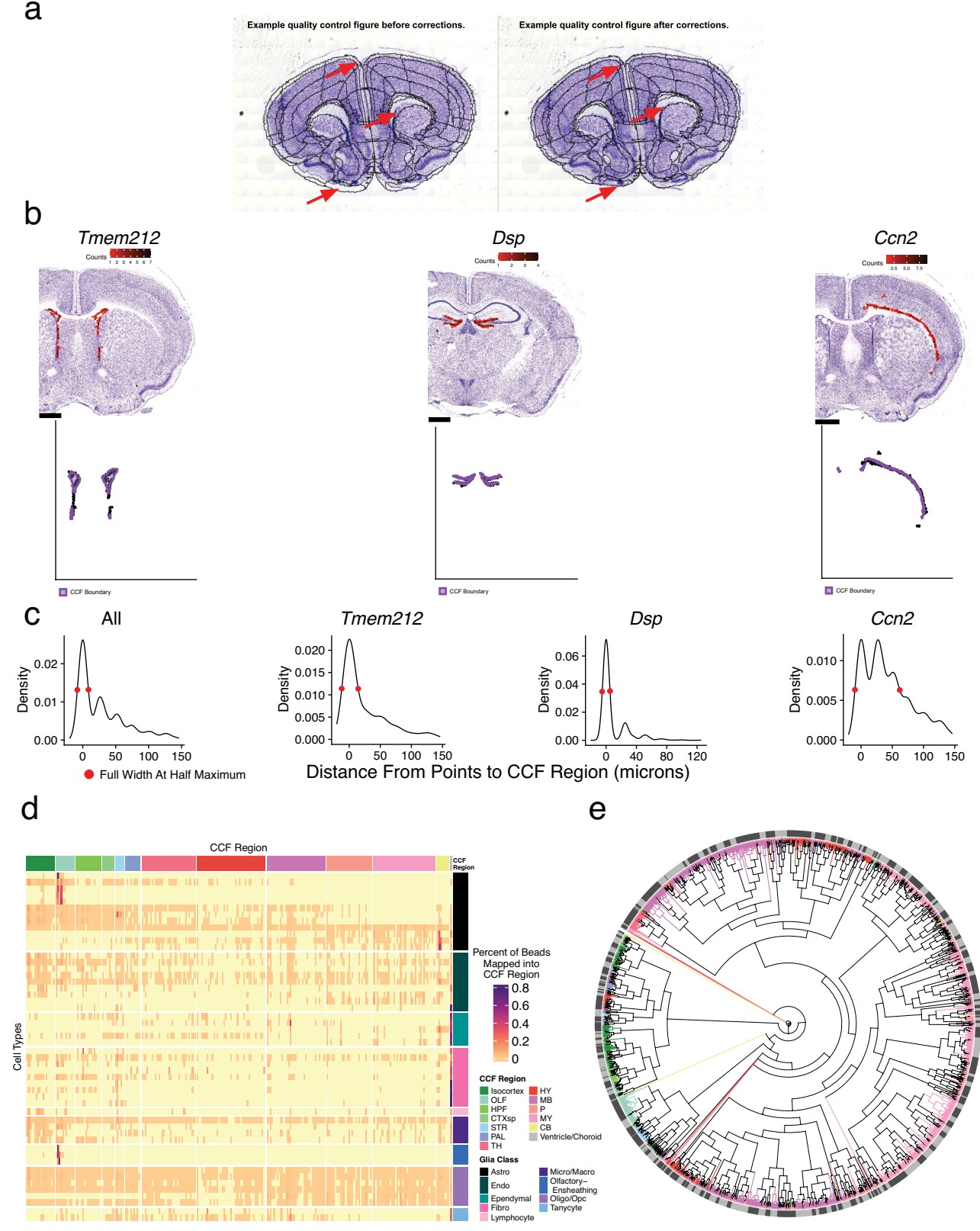

**Extended Data Fig. 2** | See next page for caption.

**Extended Data Fig. 2 | Quality control and summary statistics of CCF integration and cell type mapping. a**, Example images of adjacent Nissl sections aligned to CCF with 2D rigid transformation (wireframe outline) before (left) and after (right) correcting alignment with a 2D diffeomorphism. Red arrows point to example regions with incorrect alignment, and improvement after application of the correction. **b**, Expression of three highly specific marker genes that label the ventricular lining (*Tmem212*; 309 beads with non-zero expression), dentate gyrus granule layer (*Dsp*; 318 beads with non-zero expression), and layer 6b of isocortex (*Ccn2*; 418 beads with non-zero expression) in Slide-seq (top row); scale bar, 1 mm. Bottom row shows the positions of individual beads with expression with respect to the boundaries of the expected CCF region (purple). **c**, Density plot of the distance of each bead expressing each of the three marker genes (or all combined) shown in **b** across the corresponding Slide-seq sections. The full width half maximum of the density profile is shown. **d**, Heatmap representing the frequency of bead mappings for each glial cell type, across DeepCCF regions. **e**, Visualization of cell type dendrogram where each cluster (leaf of the tree) is coloured by their CCF region localization. The outer ring displays the boundaries of the 223 metaclusters where the alternating colours signify transitions between groups.

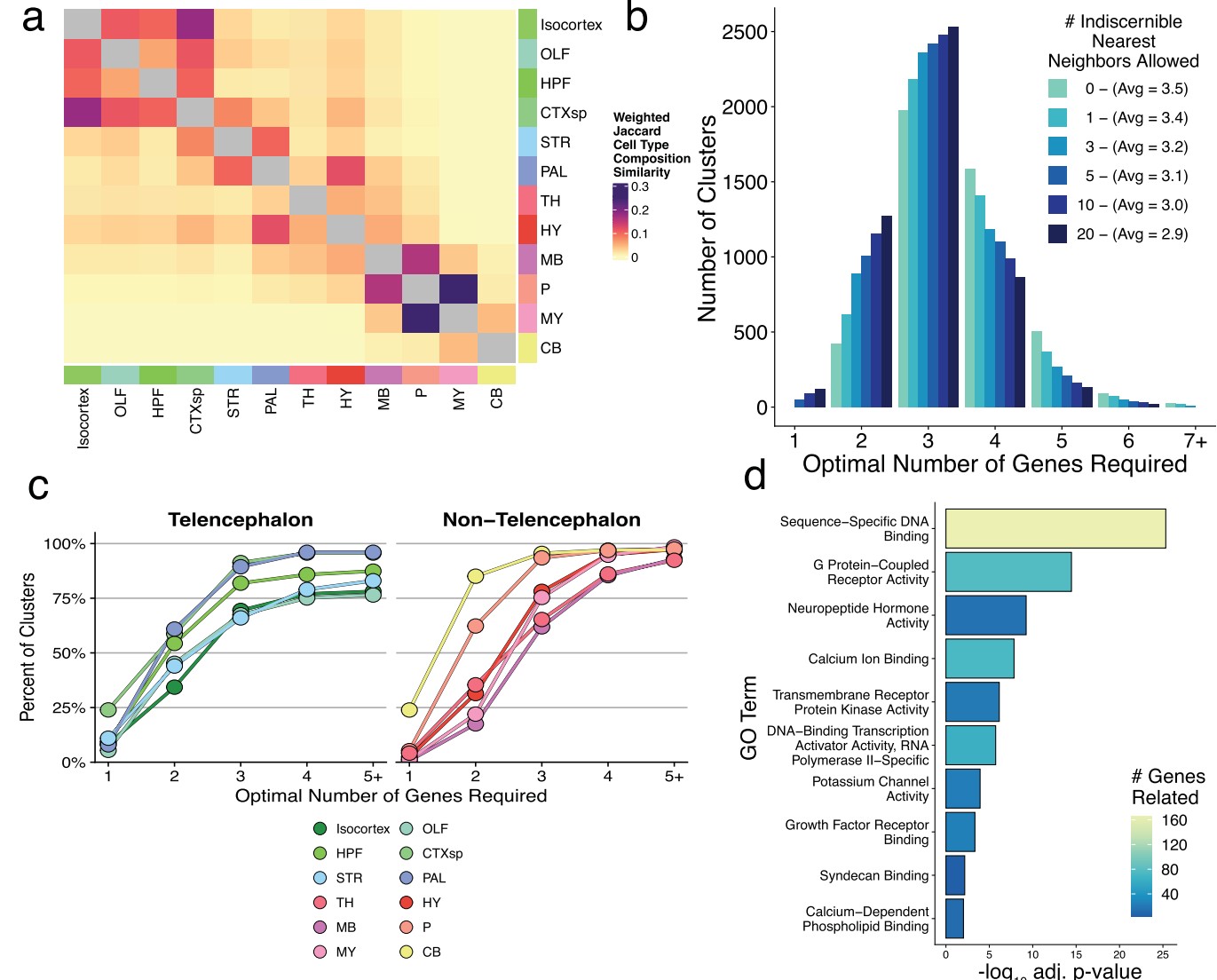

**Extended Data Fig. 3 | Extended analyses quantifying neuronal cell type diversity across brain areas. a**, Heatmap representing the weighted Jaccard similarity in cell type composition between each of the 12 main brain areas (Methods). **b**, Histogram of the minimum number of genes required to uniquely define each cell type across the nervous system. The algorithm was repeated, tolerating genes to be the same amongst different numbers of nearest cluster neighbours. **c**, Cumulative distribution plots of the number of genes needed to label each cell type, within each of the 12 major brain areas. Within each region, each of the sets of individually coloured plots denotes an algorithmic run with a different number of nearest neighbours that are tolerated as having the same gene markers (absolute number in parentheses at right). The coloured percentages denote the proportion of cell types for which the algorithm was able to find a solution. **d**, Bar plot quantifying the significantly enriched GO terms using EnrichR (hypergeometric test, p-adj <0.05) in the minimum-sized collated gene list after hierarchical reduction (Methods), coloured by the absolute number of genes related.

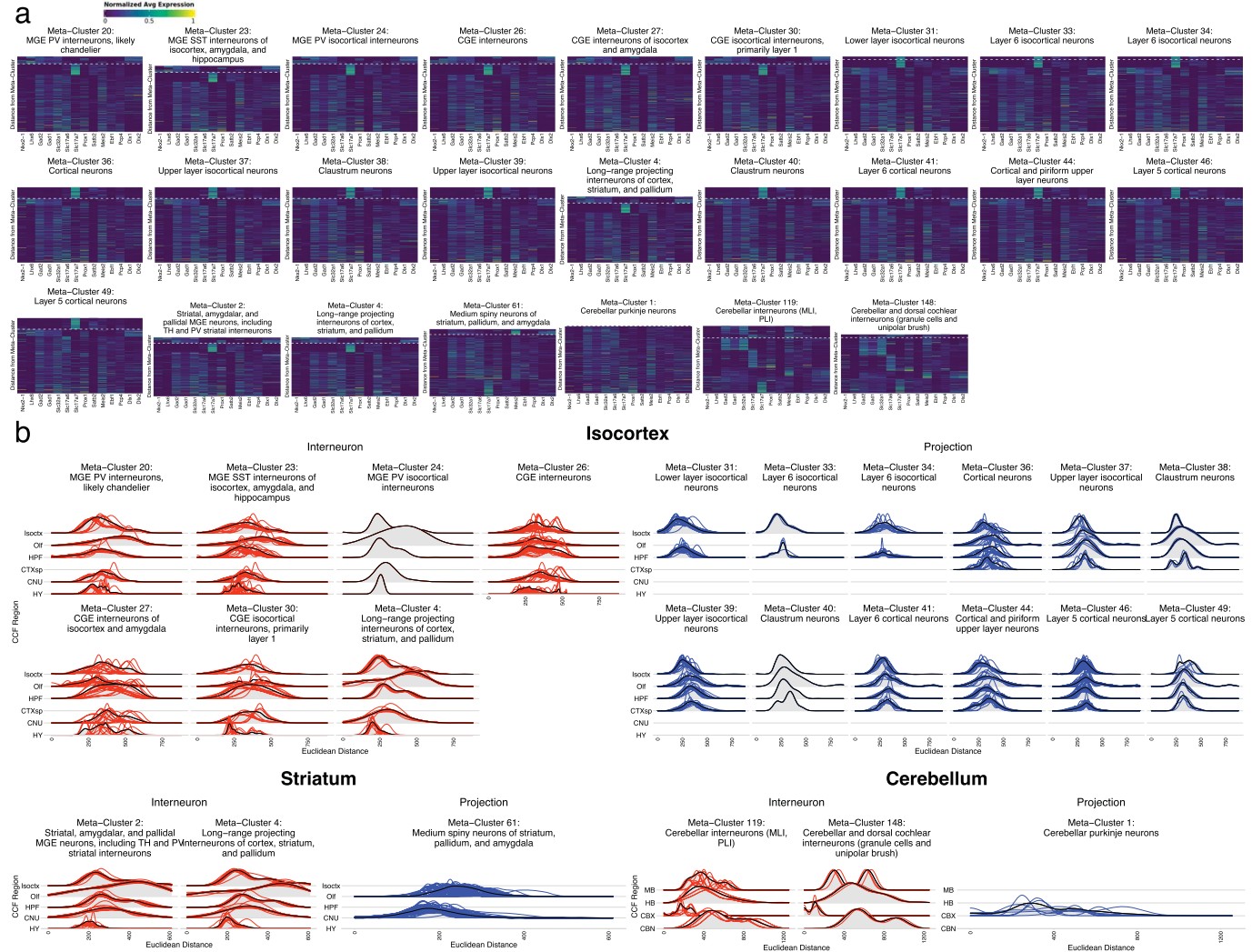

**Extended Data Fig. 4 | Extended analyses of projection and interneuron cell type relationships across regions. a**, Normalized gene expression heatmaps of the neighbouring cell types for clusters within each metacluster. The clusters are sorted by increasing distance to the index metacluster (nearest at top), with the dotted horizontal line delineating the proximate neighbourhood (Methods). Marker genes are plotted to show the calibration of neighbourhoods to include only similar cell types. **b**, Extended ridge plots depicting the transcriptomic distance from selected regions' projection and interneuron cell types to their proximate neighbourhoods, separating each neighbourhood into their CCF regions (Methods). The neurons are split into the relevant metacluster for each region.

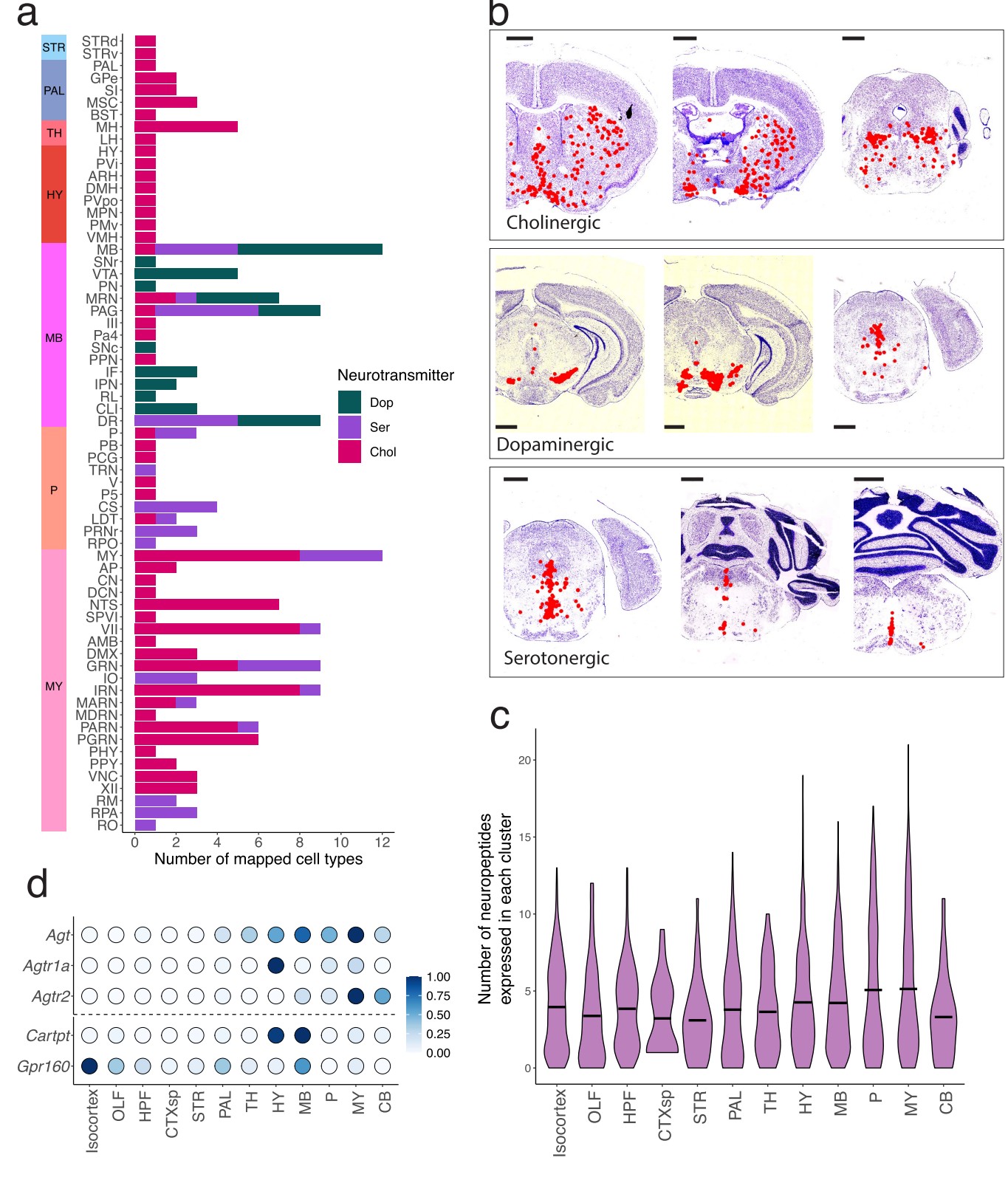

**Extended Data Fig. 5 | Extended analyses of neurotransmitter and neuropeptide usage across the brain. a**, Stacked bar plots of the number of cell types with confident mappings in each deep CCF region (conf. value > 0.3, Methods), sub-setted by neurotransmitter group. Deep CCF regions are coloured on the left by their corresponding major brain region. **b**, Representative sections showing the confident mappings of all cell types within three neurotransmitter groups (conf. value > 0.3, Methods). **c**, Violin plots of the number of neuropeptides expressed in each cluster, stratified by main brain region. **d**, Dot plot showing scaled Slide-seq counts per 10,000 of ligand-receptor pairs across main brain regions.

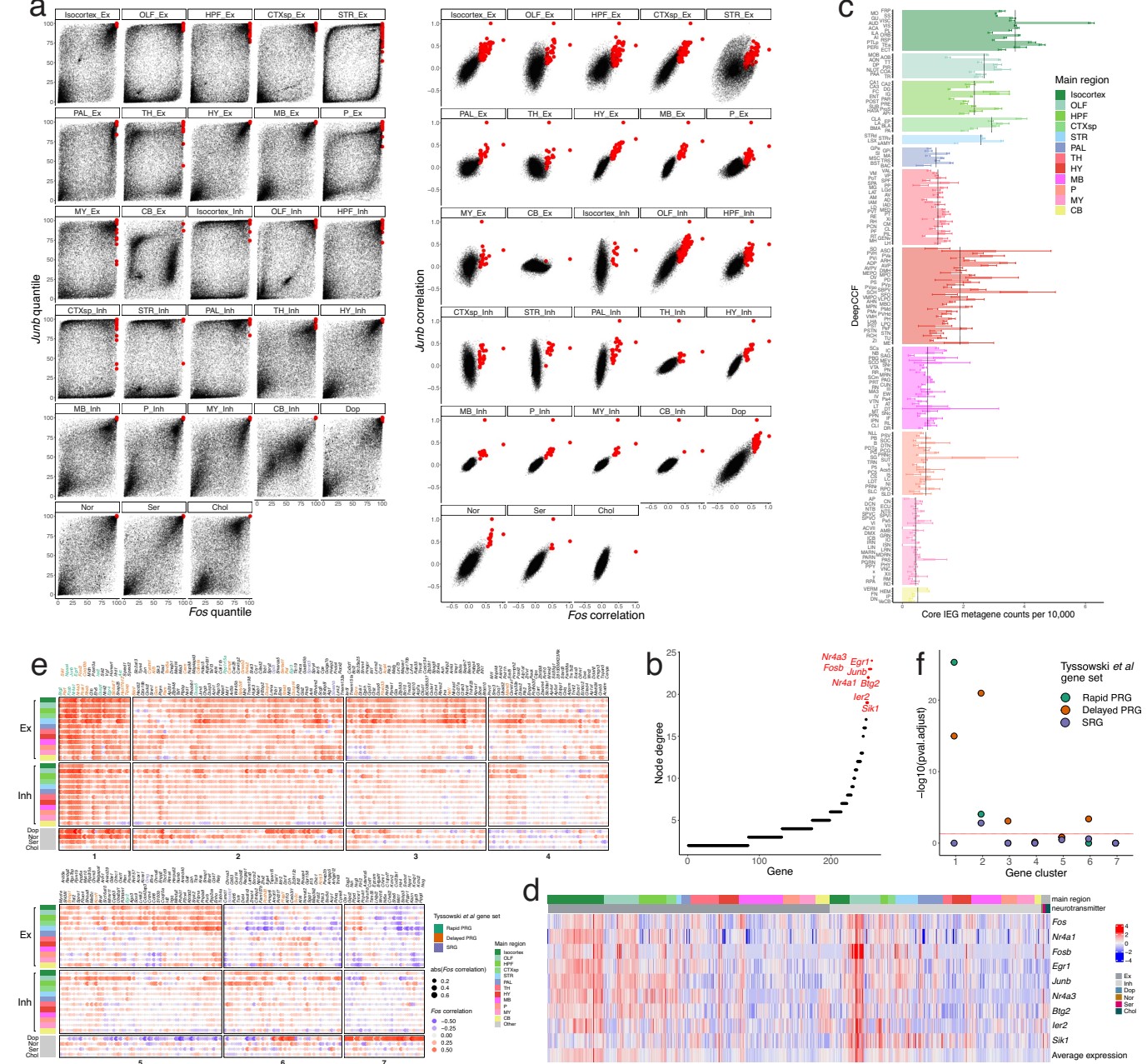

**Extended Data Fig. 6 | Extended analyses related to activity-related genes.**
**a**, Comparison of correlations (right) and quantile of correlation (left) between each gene and both *Fos* (x-axis) and *Junb*. Red dots indicate the genes that were selected as candidate ARGs. **b**, Scatter plot quantifying the node degree of each gene in the activity-regulated gene network. Genes labelled in red were selected as our core IEGs (Methods). **c**, Bar plots showing the average Slide-seq counts per 10,000 of the core IEG metagene (the eight genes highlighted in panel **b**). Error bars indicate standard error of average counts per deep CCF region and dashed black lines indicate average counts per main brain region. **d**, Scaled mean expression of the core IEGs, within each main region, separated by neurotransmitter group. **e**, Extended dot plot of correlation coefficients between *Fos* and candidate ARGs (columns) across major regions of the brain (rows). Genes are coloured by their established ARG gene set[42] identity, if applicable. Numbers at the bottom correspond to ARG cluster identities as determined by hierarchical clustering. **f**, Enrichment analysis of each candidate ARG cluster with three established ARG gene sets[42]. P-values were computed from Fisher's exact test using GeneOverlap and Bonferroni-corrected for multiple hypothesis testing (Methods). Dotted red line indicates an adjusted p-value threshold of 0.05.

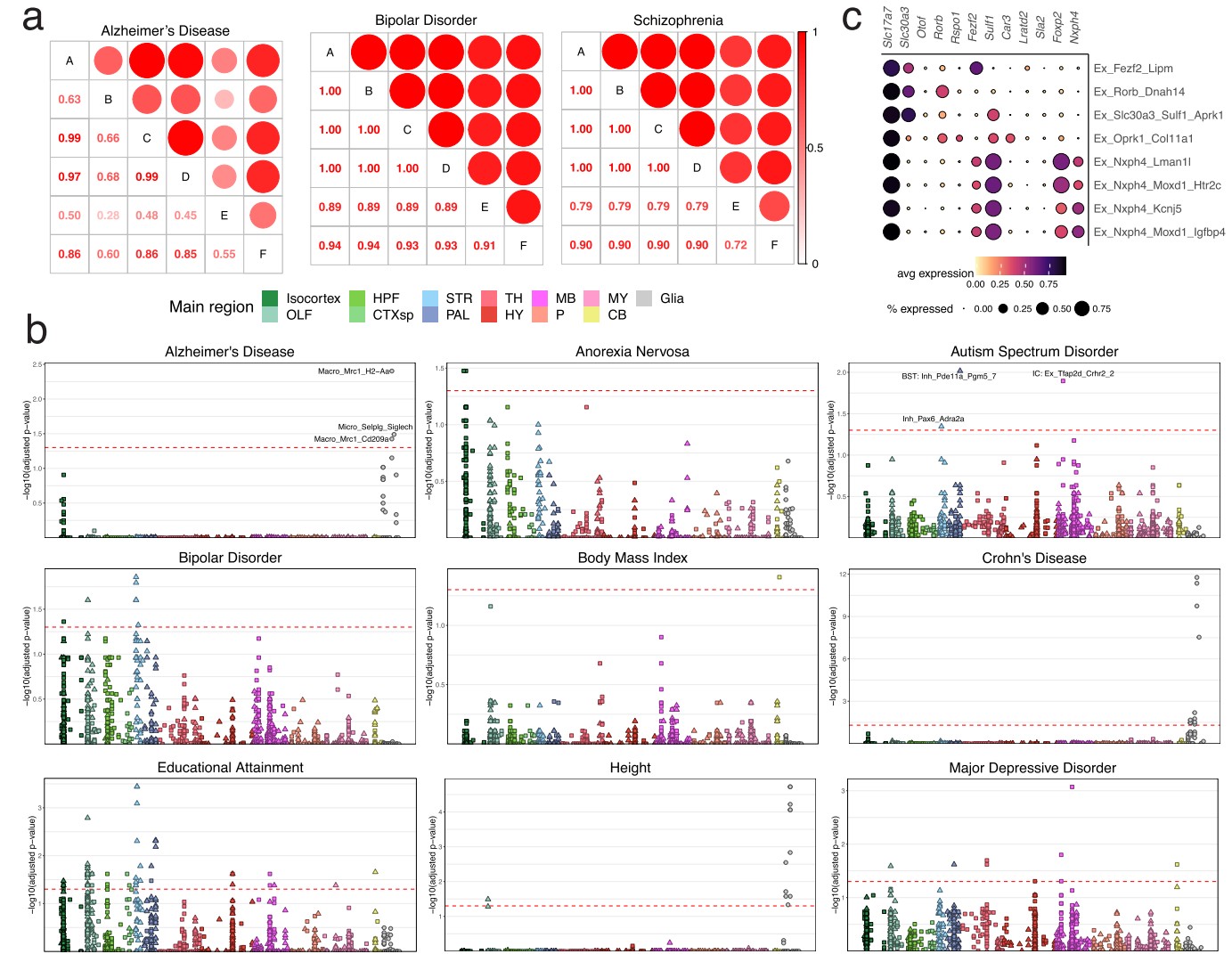

**Extended Data Fig. 7 | Extended analyses of heritability enrichment in murine brain cell types. a**, Correlation plot of p-value enrichment scores (one-sided MC test from scDRS, FDR-corrected, Methods) for each cell type across different scDRS settings: A) default parameters (Methods) B) MAGMA gene z-score > 2.5, C) control gene set size of 2,000, D) control gene set size of 500, E) input expression dataset at the single-cell level (versus pseudocells), F) adjust for cell type proportions. **b**, FDR-adjusted -log10 p-value enrichment scores for each cell type, grouped and coloured by main region, for an extended set of GWAS-measured traits. Squares and triangles denote excitatory and inhibitory clusters, respectively; glia are shown in grey on the far right of each plot. P-values computed by scDRS using a one-sided MC test. **c**, Dot plot of the expression of key cortical pyramidal cell type markers within the eight isocortical clusters that were significantly enriched (p-value < 0.05, computed by scDRS using one-sided MC test, FDR-corrected) for schizophrenia heritability.

# Reporting Summary

## Statistics

For all statistical analyses, confirm that the following items are present in the figure legend, table legend, main text, or Methods section.

| n/a | Confirmed | |
|---|---|---|
| ☐ | ☒ | The exact sample size (*n*) for each experimental group/condition, given as a discrete number and unit of measurement |
| ☐ | ☒ | A statement on whether measurements were taken from distinct samples or whether the same sample was measured repeatedly |
| ☐ | ☒ | The statistical test(s) used AND whether they are one- or two-sided<br>*Only common tests should be described solely by name; describe more complex techniques in the Methods section.* |
| ☒ | ☐ | A description of all covariates tested |
| ☐ | ☒ | A description of any assumptions or corrections, such as tests of normality and adjustment for multiple comparisons |
| ☐ | ☒ | A full description of the statistical parameters including central tendency (e.g. means) or other basic estimates (e.g. regression coefficient) AND variation (e.g. standard deviation) or associated estimates of uncertainty (e.g. confidence intervals) |
| ☐ | ☒ | For null hypothesis testing, the test statistic (e.g. *F*, *t*, *r*) with confidence intervals, effect sizes, degrees of freedom and *P* value noted<br>*Give P values as exact values whenever suitable.* |
| ☒ | ☐ | For Bayesian analysis, information on the choice of priors and Markov chain Monte Carlo settings |
| ☒ | ☐ | For hierarchical and complex designs, identification of the appropriate level for tests and full reporting of outcomes |
| ☒ | ☐ | Estimates of effect sizes (e.g. Cohen's *d*, Pearson's *r*), indicating how they were calculated |

*Our web collection on statistics for biologists contains articles on many of the points above.*

## Software and code

Policy information about availability of computer code

| Data collection | Mouse raw sequencing data was processed, aligned and converted to digital gene expression matrix format by CellRanger v.3.0.2 (available from 10x Genomics) with default settings. We used CellBender v3-alpha to remove cells contaminated with ambient RNA. For Slide-seq data, the sequenced reads were aligned to GRCm39.103 reference and processed using the Slide-seq tools pipeline (https://github.com/MacoskoLab/slideseq-tools) to generate the gene count matrix and match the bead barcode between array and sequenced reads. |
|---|---|
| Data analysis | All analysis and code used for data clustering, integration, and analysis is available at https://github.com/MacoskoLab/brain-atlas. |

For manuscripts utilizing custom algorithms or software that are central to the research but not yet described in published literature, software must be made available to editors and reviewers. We strongly encourage code deposition in a community repository (e.g. GitHub). See the Nature Portfolio guidelines for submitting code & software for further information.

## Data

Policy information about availability of data

All manuscripts must include a data availability statement. This statement should provide the following information, where applicable:

- Accession codes, unique identifiers, or web links for publicly available datasets
- A description of any restrictions on data availability
- For clinical datasets or third party data, please ensure that the statement adheres to our policy

An interactive portal is available at www.braincelldata.org, where gene expression data and spatial localizations of individual cell types can be visualized in a number

of ways. Raw single-nucleus RNA-seq and Slide-seq data is available in the NeMO Archive (www.nemoarchive.org). The snRNA-seq clustering algorithm is available in the repository www.github.com/MacoskoLab/brain-atlas. This code was built on top of several existing packages: numpy, scipy, dask, pynndescent, leidenalg, igraph, zarr, and their dependencies.

# Research involving human participants, their data, or biological material

Policy information about studies with human participants or human data. See also policy information about sex, gender (identity/presentation), and sexual orientation and race, ethnicity and racism.

| Reporting on sex and gender | Not applicable |
|---|---|
| Reporting on race, ethnicity, or other socially relevant groupings | Not applicable |
| Population characteristics | Not applicable |
| Recruitment | Not applicable |
| Ethics oversight | Not applicable |

Note that full information on the approval of the study protocol must also be provided in the manuscript.

# Field-specific reporting

Please select the one below that is the best fit for your research. If you are not sure, read the appropriate sections before making your selection.

☒ Life sciences ☐ Behavioural & social sciences ☐ Ecological, evolutionary & environmental sciences

For a reference copy of the document with all sections, see nature.com/documents/nr-reporting-summary-flat.pdf

# Life sciences study design

All studies must disclose on these points even when the disclosure is negative.

| Sample size | No statistical methods were used to predetermine sample size for animals used to generate transcriptomic data; sample sizes were selected to ensure multiple replicates, in each region, for each sex. |
|---|---|
| Data exclusions | For transcriptomic data, pre-established exclusion criteria of a minimum of 500 UMI/nucleus and a maximum of 1% mitochondrial nUMIswere used, as these were standard thresholds used in previous studies. |
| Replication | Clusters generated from the transcriptomic data were examined to ensure that multiple technical replicates contributed to individual clusters. Analyses relevant to replication can be found in Extended Data Figure 1. |
| Randomization | No differential experimental treatments were applied to individuals or samples in this study. |
| Blinding | Blinding was not relevant in this study as no differential experimental treatments were applied. |

# Reporting for specific materials, systems and methods

We require information from authors about some types of materials, experimental systems and methods used in many studies. Here, indicate whether each material, system or method listed is relevant to your study. If you are not sure if a list item applies to your research, read the appropriate section before selecting a response.

## Materials & experimental systems

| n/a | Involved in the study |
|---|---|
| ☒ | ☐ Antibodies |
| ☒ | ☐ Eukaryotic cell lines |
| ☒ | ☐ Palaeontology and archaeology |
| ☐ | ☒ Animals and other organisms |
| ☒ | ☐ Clinical data |
| ☒ | ☐ Dual use research of concern |
| ☒ | ☐ Plants |

## Methods

| n/a | Involved in the study |
|---|---|
| ☒ | ☐ ChIP-seq |
| ☒ | ☐ Flow cytometry |
| ☒ | ☐ MRI-based neuroimaging |

# Animals and other research organisms

Policy information about studies involving animals; ARRIVE guidelines recommended for reporting animal research, and Sex and Gender in Research

| | |
|---|---|
| Laboratory animals | Mouse transcriptomic data was generated from adult female and adult male mice (60 days old; C57BL/6J, Jackson Labs). Specific data about the mice used can be found in Extended Data Figure 1. |
| Wild animals | No wild animals were used in the study. |
| Reporting on sex | Both male and female mice were jointly used in the generation of the single cell clustering data. |
| Field-collected samples | No field-collected samples were used in the study. |
| Ethics oversight | All procedures involving animals were conducted in accordance with the US National Institutes of Health Guide for the Care and Use of Laboratory Animals under protocol number 1115-111-18 and approved by the Massachusetts Institute of Technology Committee on Animal Care. |

Note that full information on the approval of the study protocol must also be provided in the manuscript.

# Plants

| | |
|---|---|
| Seed stocks | N/A |
| Novel plant genotypes | N/A |
| Authentication | N/A |

