## [Peer Review File · Nature]

Manuscript Title: The molecular cytoarchitecture of the adult mouse brain

Reviewer Comments & Author Rebuttals

Reviewer Reports on the Initial Version:

Referees' comments:

Referee #1:

Remarks to the Author:

In Langlieb and Sachdev et al from the Macosko and Chen groups, the authors present a tour-de-force single nucleus and spatial transcriptional atlas of the adult mouse brain, producing unprecedented cell type granularity with the ability to position these cell types within well-defined structures in the brain. The work not only provides an incredibly valuable resource, but the analyses performed take advantage of the vast scope of the dataset, enabling novel observations that would not have previously been possible. These observations, including minimal gene sets for highly-specific neuronal subtype targeting will undoubtedly be valuable for the generation of genetic tools, or in providing additional context to other studies. This also applies to the other major components of the work that include the neuropeptide analysis and activity based analysis. The integration with GWAS hits also adds value by first identifying previously established cell type associations but then taking it a step farther in leveraging the granularity and spatial components of the work. Overall the molecular methodologies deployed are now well-established by the groups and were applied with the same rigor as prior studies, just at a far greater scale. Similar to the molecular methodologies, many of the initial computational analyses and approaches to define clusters and granularity have been deployed previously – though again at a much larger scale in this work. The robustness of these analysis strategies is apparent when considering how they enable a much more granular view of cell types when applied to a much larger dataset. The subsequent, novel analyses are well thought out and are clever ways to interrogate the wealth of data to uncover meaningful insight into neuronal diversity and how these granular subtypes interact with one another.

The majority of cells profiled were neuronal with far fewer glia (after revisiting the clusters as capped at 2k cells, so it is not clear if this is the case, this further emphasizes some more summary level info in the main text and figure) – generally fewer than I would expect, though I am not versed in the composition of each of the 92 anatomical locations profiled. A brief sentence relating the proportions of major cell type classes to what is known from imaging datasets would be helpful as well as the reasoning for why neurons are so highly enriched (which is good given the diversity of cell types). It would also be nice to see a little more analysis on the glial populations, even cursory. The data are there and many groups that focus on glia would benefit.

A note on the snRNAseq data - the doublet removal appears only at the level of cluster QC where very little detail is provided. At this scale of cell counts, one would expect doublets to form their own clusters, and the analysis performed is important. Currently the methods text describes removing clusters with co-expression of distinct marker genes then provides only one such example. One problem with this approach is that it cannot be used in identifying doublet clusters where markers are not well known. Furthermore, additional detail must be provided here: what were the co expression markers that were used? What clusters were removed? Of all QC passing cells, what percentage of them are in doublet clusters and how does that percentage relate to the cell loading onto the instrument which can be used as an expected doublet percentage? And given the number of runs - a valuable confirmation would be relating the percent of cells in doublet clusters versus the number of cell profiles obtained from each individual instrument channel.

Page 2, 2nd to last paragraph: The alignment to CCF structures is a very important component of this work, and to those that have used the Allen CCF it is intuitive; however, a little more description would benefit the broader

readership. The way it is written now terms like CCF region and structure are used without definitions – ‘main regions’ are also used later in the text. I appreciate the brevity in the current text, but adding a little more context here would benefit the manuscript particularly for laying out the terms used throughout the rest of the text.

Most of the key cell type determining genes are transcription factors. Many TFs are known to have burst expression that produces stable protein and does not need to be continually expressed. While considering the expression dynamics of every TF (which is also likely varied in different cell types, and also largely unknown) is not possible at this time – it could be worth noting something on this; basically, relaying a caveat that some of these clusters may reflect a certain state versus cell type. This caveat is especially relevant to the design of future genetic tools. It is worth noting though, that the spatial mapping component lends a great deal of confidence to distinct cell types, versus a fluctuating state captures at any given time – at least for those that are highly specific to certain CCF structures – taking that into account as an added layer of design in genetic tools could be worth noting.

“We also discovered that most cell types had more than one distinct minimally sized gene list.” - this sentence threw me for a bit of a loop - I had to go back and re-read the previous paragraphs. Some more description here would help it flow - on first read the sentence seems contradictory since one would assume a single minimally sized gene list would be identified for each individual cell type or cluster. Rewording this would help clarify or possibly setting up this a bit more clearly in the prior paragraph - indicating how multiple sets can be identified for a given cell type.

Fig 3 a & b - some of the text is just way too small. I believe minimum allowable font size for most journals is 6pt. I know it will be difficult to fit all of the labels, but adhering to ≥ 6 pt will make it far more interpretable. Also panel b is super cool - just blow the whole thing up to be larger until the label text is the right size - it deserves the space.

In the NP section - last paragraph of page 5 - some of the verbage switched from NP expressing and NP releasing. As far as I can tell there is no measurement of release in the data produced, just expression and then assumed release. While it is heavily assumed, the text should stick to NP expression versus release. This also goes to the next paragraph stating many NPs are sensed by cell types, when the data is that the NP receptor is expressed, which again leads to the assumption it has the ability to sense that NP - it is still not direct evidence. The analysis itself in this section is very interesting and illustrates the power of the datasets.

Figure 4a is hard to make sense of and draw any conclusions. It suffers from the classic hairball structure, and the only take home is the large gene names. Is there any way to summarize these data in a different plot? This could be included in the supplement, though I still am not sure it adds much. I like the concept of the analysis, but really need another visualization - perhaps splitting into the two sets and reporting summary level values? Just something more digestible. Panels b-d are great.

Page 8, 2nd Para, Line 4: ‘single cell datasets’ -> ‘single-cell datasets’

Just an idea: It is clear that for datasets this massive that traditional tSNE or UMAP etc... fail. I am not suggesting redoing the figure - it is the best we can really do at this time. That being said - have you tried aggregating RNA profiles for each cluster then performing UMAP or tSNE on that? Perhaps then plotting each point sized by the log10 cell count? 5,000 points versus 1.2M (which is only a subset!) would likely allow the visualization tool ‘more space’. Solving the single-cell visualization problem is beyond the scope of this paper, but the above suggestion would probably be pretty easy to do. Fig 2b captures this same theme, though has a whole additional layer of information which is pretty powerful.

Referee #2:

Remarks to the Author:

Langlieb et al. present a comprehensive atlas of the mouse brain, using single-nucleus RNA sequencing and spatial transcriptomics. The combination of snRNA-seq and Slide-seq is stunning and will be a fantastic resource for mouse brain research. Having the spatial data is extremely important, as it potentially helps interpret the much more abstract single-cell data. It also helps align the molecular cell type atlas with past and future imaging data.

* Overall impression

The technical aspects are excellent and as a resource, the results will be invaluable. I have only minor questions and requests (see below). The biological analyses were fine for what they were, with some caveats (below). For example, the discovery of combinatorial markers of cell types will be a valuable resource for experimentalists to access defined cell types, but the discussion of those markers was perhaps unnecessarily extensive as they provided little insight into the brain's organisation.

What I found lacking however, was any kind of discussion of organising principles. One gets the impression from the current manuscript that the brain is just a bag of 4000+ neuron types and tens of glial types. But having a combined single-cell and spatial dataset is a great opportunity to discover how the brain is organised. For example, it would probably be useful to define one or more intermediate levels of analysis, e.g. "meta clusters" or clades in a dendrogram. That may reveal broader patterns of neural cell types, e.g. beyond telencephalic vs non-telencephalic astrocytes, or the distribution of telencephalic excitatory neurons outside the cortex. It may also reveal classes of cells that migrate during development, e.g. ganglionic eminence-derived cortical interneurons, and their possible migration to other target structures e.g. thalamus.

More broadly, are there any organising principles of brain cell types? Do they reflect development? Evolution? function (e.g. neurotransmitter)? Something else? Are glia and neurons organised according to similar patterns, or not? Were brain cell types organised hierarchically? Combinatorially? Was the organising principle different in e.g. telencephalon vs brainstem? What was the relationship between cortical areas? Do any cortical areas stand out with having unique cell types?

The spatial images shown reveal stunningly precise localisation of some neuron types, but I would have loved a more systematic analysis. For example, the brainstem is traditionally considered to consist of "nuclei" - does this hold true? E.g. are most brain stem neuron types confined to precisely localised nuclei, or are they more dispersed? Are there different classes of neurons - layered (as in cortex), condensed (nuclei), dispersed, ...? If so do the classes have anything in common?

Integrating spatial and single-cell data presents an opportunity of unprecedented analysis of cortical neurons. Could you computationally flatten the cortical sheet and ask about the organising principles of neurons across the sheet? There is ample evidence for e.g. spatial gene expression gradients and differences in cell type composition between cortical areas, but you are in a position to almost have the final word. How are cortical neurons organised spatially across the cortical sheet? Are there systematic groups of such cell types? Are the layer-specific neurons organised according to the same or different principles? Do neocortical types extend into piriform or entorhinal cortex, or the amygdala? If so, which ones do and which ones don't. Etc.

Another interesting aspect concerns sex. Were you powered to study sex differences, and did you find any? Were there sex differences in cell types, in cell type proportions, or in specific genes expressed in cell types?

* Technical issues

On p. 10 "Regional dissections": what were the precise extents of the anatomical regions, and how confident were the borders of the dissected tissue blocks? In other words, to what extent should we expect a region to also contain contaminating cells from adjacent regions, or to potentially miss cells located near its borders? For example, the photo in Fig 1a suggests that dissections deviated quite strongly from neuroanatomical borders. If the photo

documentation is available, it would be great if the authors would share it e.g. on a companion web site. Alternatively, polygon coordinates could be added to the supplementary table that defines dissections, so that they can be overlaid on the anatomical coordinate framework.

The tSNEs in Fig 1 use parameters that lead to filling the unit circle, again giving the impression of a bag of cell types. It would probably be informative to increase the exaggeration parameter so as to yield a stronger global structure (this can be done e.g. with OpenTSNE; see also the paper "The art of using tSNE for single-cell transcriptomics", which has good heuristics for the parameters). UMAP will typically have a similar effect. Perhaps this would help define additional intermediate levels of analysis mentioned above.

For Extended Fig 1g, a table of numbers would be more informative (log-scaled bars are very hard to read anyway). Were there exactly 1 microglia and 1 OPC cluster, i.e. those zero-height bars are actually informative?

* Improved analysis of non-neuronal cell types.

Were there ~20 endothelial cell types? What were they? Did you identify any choroid plexus-specific endothelial cells, with possible specialised function in producing CSF fluid? To my knowledge, previous work in mouse has failed to identify convincing region-specific endothelial cells. Did you find any? Or do all the ~20 clusters correspond to the arterial-capillary-venous gradient?

Extended Fig 2d is hard to read since there is no annotation on the vertical axis. There seem to be >50 glial types, but what are they? Please indicate what they are (ependymal, astrocyte, ...) . There seems to be quite a lot of region-specific glia. What were they?

Did you find only one kind of OPC? That would agree with previous work in mice, but disagree with recent work in the human brain (e.g. Siletti et al. bioRxiv 2022). It would be potential evidence of a potential human vs mouse specialisation!

You seem to find multiple types of macrophages - what were they? E.g. distinct macrophages of different brain borders (meninges, vascular, ...)?

Were there no other immune cells? That in itself would be very interesting, since there is a common belief that there must be some lymphoid cells in the normal brain even if they are rare. You could put a number on that.

You seem to find a large number of fibroblasts types — interesting! What were they? Meninges layers? Did you remove meninges, or not (please add to Methods)? If you have meninges, do you see any evidence of the recently described SLYM layer? It has been very controversial. Were the meninges different dorsally vs ventrally, or anterior vs posterior? This is relevant because meningioma subtypes have characteristic spatial patterns. Did you find specialised perivascular fibroblasts?

Author Rebuttals to Initial Comments:

Referee Responses, Langlieb, Sachdev *et al*, 2023

We very much appreciate the feedback from the referees; our responses to them over the past several months have greatly strengthened the quality of our manuscript. In brief, the main additions to our revised manuscript include:

1. Hierarchical clustering of the cell populations, and their aggregation into a small number of 224 metaclusters that have been annotated with more information about their spatial origins and biological identities. Details can be found on page 12.
2. An analysis of the distribution of related cell populations throughout the brain. Specifically, we utilized the dendrogram produced by our hierarchical clustering to define neighborhoods around each cell type (clusters with molecular similarity to the index cluster). We then asked how these neighborhoods of each cell type are distributed across brain areas. We found that interneurons–neuron populations that signal locally, rather than across long distances–have more widely dispersed neighborhoods than projection neurons. In other words, projection neurons are more unique to the structures they inhabit, while interneurons share molecular features with each other across far more diverse brain structures. Our discussion of these new analyses begins on page 13.
3. A more detailed analysis and discussion of non-neuronal cells. Details can be found on pages 2 and 19-24 of this document.

Below, we delineate our responses to each comment.

Referee #1

In Langlieb and Sachdev et al from the Macosko and Chen groups, the authors present a tour-de-force single nucleus and spatial transcriptional atlas of the adult mouse brain, producing unprecedented cell type granularity with the ability to position these cell types within well-defined structures in the brain. The work not only provides an incredibly valuable resource, but the analyses performed take advantage of the vast scope of the dataset, enabling novel observations that would not have previously been possible. These observations, including minimal gene sets for highly-specific neuronal subtype targeting will undoubtedly be valuable for the generation of genetic tools, or in providing additional context to other studies. This also applies to the other major components of the work that include the neuropeptide analysis and activity based analysis. The integration with GWAS hits also adds value by first identifying previously established cell type associations but then taking it a step farther in leveraging the granularity and spatial components of the work. Overall the molecular methodologies deployed are now well-established by the groups and were applied with the same rigor as prior studies, just at a far greater scale. Similar to the molecular methodologies, many of the initial computational analyses and approaches to define clusters and granularity have been deployed previously – though again at a much larger scale in this work. The robustness of these analysis strategies is apparent when considering how they enable a much more granular view of cell types when applied to a much larger dataset. The subsequent, novel analyses are well thought out and are clever ways to interrogate the wealth of data to uncover meaningful insight into neuronal diversity and how these granular subtypes interact with one another.

We appreciate the reviewer's positive feedback on our work. The generation of these datasets was an enormous effort, spanning six years and many individuals in our labs. We hope that

these data and our initial analyses of them are a valuable reference for the neuroscience community for years to come.

The majority of cells profiled were neuronal with far fewer glia (after revisiting the clusters capped at 2k cells, so it is not clear if this is the case, this further emphasizes some more summary level info in the main text and figure) – generally fewer than I would expect, though I am not versed in the composition of each of the 92 anatomical locations profiled. A brief sentence relating the proportions of major cell type classes to what is known from imaging datasets would be helpful as well as the reasoning for why neurons are so highly enriched (which is good given the diversity of cell types).

We apologize this was not clear in the text: we downsampled the glia (and other very large clusters) in order to show a more balanced representation of cell types. This downsampling was performed exclusively to generate our t-SNE, and construct our cell type dendrogram (a new addition since revision, see below), but it was not used in any other analysis. Extended Data Figs. 1d,h, for example, show each cell class's representation within our full, non-downsampled dataset. In our revision, we provided more clarification in the Methods section of the text about where our downsampled cells were used. Specifically, we mention our downsampled cells in the sections entitled "Visualization of clusters" and "Constructing Paris dendrogram and aggregation into groups".

It would also be nice to see a little more analysis on the glial populations, even cursory. The data are there and many groups that focus on glia would benefit.

We appreciate both reviewers' general encouragement to pursue a more detailed annotation and analysis of the glial populations. We admit that our fascination with the extraordinary diversity in the neurons occupied the great majority of our attention, and we somewhat neglected the glia in our initial submission.

We have revised Extended Data Fig. 2d to more clearly capture the spatial localizations of individual glial populations, grouped by broader classes. In addition, we have added a paragraph about our glial findings at the beginning of the results section of the main text, reproduced below:

"Most glial populations were distributed across large neuroanatomical boundaries (telencephalon, mesencephalon, and rhombencephalon), indicating that, relative to neurons, regional gene expression differences amongst glial populations were small (Extended Data Fig. 2d). A single oligodendrocyte precursor (OPC) cluster was identified, in contrast to a recent report of additional OPC subspecialization in humans¹. The glial clusters with regional segmentation included astrocytes, which divided into olfactory-specific, telencephalic, and non-telencephalic populations, as well as a cerebellum-specific population (the Bergmann glia). Amongst our endothelial cell populations, we identified populations preferentially localized to the choroid plexus (Extended Data Fig. 2e). Additional regionally localized glial populations included the olfactory ensheathing neurons, identified by their expression of the known marker homeobox genes *Alx3* and *Alx4*², and hypothalamic tanycytes, which uniquely express *Rax*³."

A note on the snRNAseq data - the doublet removal appears only at the level of cluster QC where very little detail is provided. At this scale of cell counts, one would expect doublets to form their own clusters, and the analysis performed is important. Currently the methods text describes removing clusters with co-expression of distinct marker genes then provides only one such example. One problem with this approach is that it cannot be used in identifying doublet clusters where markers are not well known. Furthermore, additional detail must be provided here: what were the co-expression markers that were used? What clusters were removed? Of all QC passing cells, what percentage of them are in doublet clusters and how does that percentage relate to the cell loading onto the instrument which can be used as an expected doublet percentage? And given the number of runs - a valuable confirmation would be relating the percent of cells in doublet clusters versus the number of cell profiles obtained from each individual instrument channel.

We appreciate the reviewer's attention to the details of our dataset QC, and the specific suggestions for how to report, and justify, doublet removal.

We have now included additional information about how doublets were identified and removed in the Methods section of the manuscript. The full set of co-expression markers used to identify doublets is now provided in Supplemental Table 2.

To address the reviewer's point about identifying doublet clusters with unknown markers, we demonstrate that all of our clusters (prior to cluster removal for any QC reason) express marker genes from a known cell class:

Since all clusters can be assigned to a cell class, we believe it is very unlikely that an entirely unknown cell class has been missed.

We have also provided information about the number of cells and clusters removed at each stage of QC, in the form of Sankey diagrams in Extended Fig. 1a. In the diagram, the doublet removal step occurs after the snRNA-seq dataset has gone through an initial round of low-quality cell removal (based upon nUMI and percent mitochondrial thresholds—see below), as

well as hierarchical clustering and cell type annotation. At this doublet removal stage, we removed 7.4% of QC-passing cells:

This corresponds to 8.7% of all clusters:

To demonstrate our doublet removal percentages are concordant with the predicted 10X snRNA-seq multiplet rate of cells loaded, we followed the reviewer’s suggestion and plotted the proportion of doublets removed as a function of 10X library size (binned into six groups):

We see the expected positive relationship, with doublet percentages roughly concordant with 10X multiplet rates. i.e. 10X predicts a library with ~10k recovered cells to have a multiplet rate of ~8%, and, on average, we see a similar doublet rate in our libraries of that size.

Page 2, 2nd to last paragraph: The alignment to CCF structures is a very important component of this work, and to those that have used the Allen CCF it is intuitive; however, a little more description would benefit the broader readership. The way it is written now terms like CCF region and structure are used without definitions – ‘main regions’ are also used later in the text. I appreciate the brevity in the current text, but adding a little more context here would benefit the manuscript particularly for laying out the terms used throughout the rest of the text.

We appreciate the reviewer’s suggestions to make our manuscript more accessible to the general scientific community. In our revision, we now include the following additional detail regarding CCF alignment and CCF regional definitions used with the accompanying Supplemental table 4:

“We aligned the sequencing-generated Slide-seq images to images of adjacent histological sections, which are rich in neuroanatomical detail. To assign beads to specific neuroanatomical atlas structures, we aligned the adjacent histological sections to the Allen Common Coordinate Framework⁴ (CCF Framework) (Methods, Extended Data Fig. 2a). This CCF provides hierarchical regional definitions, allowing us to tag each Slide-seq bead with a “Main Region”--one of twelve large structural components of the brain (e.g. isocortex, striatum, etc)--as well as more fine-grained regional definitions, which we call “DeepCCF” structures (listed in Supplemental Table 4).”

Most of the key cell type determining genes are transcription factors. Many TFs are known to have burst expression that produces stable protein and does not need to be continually expressed. While considering the expression dynamics of every TF (which is also likely varied in different cell types, and also largely unknown) is not possible at this time – it could be worth noting something on this; basically, relaying a caveat that some of these clusters may reflect a certain state versus cell type. This caveat is especially relevant to the design of future genetic tools. It is worth noting though, that the spatial mapping component lends a great deal of confidence to distinct cell types, versus a fluctuating state captures at any given time – at least for those that are highly specific to certain CCF structures – taking that into account as an added layer of design in genetic tools could be worth noting.

We thank the reviewer for these thoughtful suggestions regarding the interpretation of our marker analysis. We have included these ideas in the discussion as reproduced below:

“Interestingly, we noted a large enrichment of TFs amongst the list of genes that most concisely define individual cell types. Combinatorial TF expression is a recurring theme, across CNS structures, in the neurodevelopmental specification of diverse neural cell types⁵. While it is clear in our data that many of these TF combinations represent fixed cell type specifications (based upon our knowledge of how certain TF control development in particular brain areas), additional single-cell data—acquired at different times of day, and in response to different environmental challenges—will be needed to understand which of these clusters represent populations fixed in development, and which are more mutable in response to challenges experienced in adulthood.”

“We also discovered that most cell types had more than one distinct minimally sized gene list.” - this sentence threw me for a bit of a loop - I had to go back and re-read the previous paragraphs. Some more description here would help it flow - on first read the sentence seems contradictory since one would assume a single minimally sized gene list would be identified for each individual cell type or cluster. Rewording this would help clarify or possibly setting up this a bit more clearly in the prior paragraph - indicating how multiple sets can be identified for a given cell type.

We thank the reviewer for drawing attention to this confusing sentence. We reworded that paragraph, giving an example cell type Ex_Pitx2_Zbtb7c_3 along with a few of its (optimally) equally-sized gene sets, to help illustrate the point. We additionally drew greater attention to the ultimate goal of this analysis: understanding the degree of gene redundancy within solutions and whether these repeated genes are enriched within any Gene Ontology (GO) classification. Overall, our revised manuscript now allocates less bandwidth to the set-cover solutions, instead highlighting our new analyses on the spatial localization of related cell types.

Fig 3 a & b - some of the text is just way too small. I believe the minimum allowable font size for most journals is 6pt. I know it will be difficult to fit all of the labels, but adhering to >=6pt will make it far more interpretable. Also panel b is super cool - just blow the whole thing up to be larger until the label text is the right size - it deserves the space.

We thank the reviewer for pointing out the sections of Figure 3 with small font size, and we agree that the panel sizes should be significantly increased. We have modified our entire figure and rearranged the layout to maximize interpretability, reproduced below:

In the NP section - last paragraph of page 5 - some of the verbage switched from NP expressing and NP releasing. As far as I can tell there is no measurement of release in the data produced, just expression and then assumed release. While it is heavily assumed, the text should stick to NP expression versus release. This also goes to the next paragraph stating many NPs are sensed by cell types, when the data is that the NP receptor is expressed, which again leads to the assumption it has the ability to sense that NP - it is still not direct evidence. The analysis itself in this section is very interesting and illustrates the power of the datasets.

We appreciate the reviewer's suggestion to reconsider the vocabulary used when discussing neuropeptide usage. As recommended, we have modified all phrases with "NP releasing" to "NP

expressing” and “NP sensing” to “GPCR/receptor expressing” to reflect the exact information we have about these neuropeptides in our datasets.

Figure 4a is hard to make sense of and draw any conclusions. It suffers from the classic hairball structure, and the only take home is the large gene names. Is there any way to summarize these data in a different plot? This could be included in the supplement, though I still am not sure it adds much. I like the concept of the analysis, but really need another visualization - perhaps splitting into the two sets and reporting summary level values? Just something more digestible. Panels b-d are great.

We appreciate the reviewer’s points about the network visualization, and agree that the plot doesn’t provide the audience with a clear and direct interpretation of the data. To avoid any confusion, we have removed the network from Figure 4. We still use the set of eight core IEGs throughout our analysis, which were identified as the most highly connected genes in the *Fos* correlation network. Therefore, we have included a supplementary plot (Extended Data Fig. 6b) highlighting these genes:

Page 8, 2nd Para, Line 4: ‘single cell datasets’ -> ‘single-cell datasets’

This correction has been made.

Just an idea: It is clear that for datasets this massive that traditional tSNE or UMAP etc... fail. I am not suggesting redoing the figure - it is the best we can really do at this time. That being said - have you tried aggregating RNA profiles for each cluster then performing UMAP or tSNE on that? Perhaps then plotting each point sized by the log10 cell count? 5,000 points versus 1.2M (which is only a subset!) would likely allow the visualization tool ‘more space’. Solving the single-cell visualization problem is beyond the scope of this paper, but the above suggestion would probably be pretty easy to do. Fig 2b captures this same theme, though has a whole additional layer of information which is pretty powerful.

We appreciate the reviewer’s helpful suggestions on how to improve our t-SNE visualization. After aggregating our cells into a pseudobulk representation, we found that on a global scale, the t-SNE was certainly able to afford more distance between major cell classes and spatial

localizations. However, the distance between local neighborhoods is less convincing, particularly given the nature of our highly diverse neuronal cell types:

Therefore, we decided to generate an improved t-SNE at single-cell resolution, following some of Reviewer #2's suggestions. Using OpenTSNE⁶, we sought to find optimal t-SNE settings that maximized both local and global structure, based on the guidelines laid out by the authors of "The art of using t-SNE for single-cell transcriptomics"⁷. Namely, we performed a sweep of parameters, varying both the perplexity and exaggeration factor of our t-SNE coordinates. To determine which t-SNE settings were most optimal, we compared the pairwise euclidean distances of cell type t-SNE centroids (D1), to the pairwise euclidean distances of cell type %expression across highly variable genes (D2). We computed two scores: 1) the Spearman correlation of D1 vs. D2, and 2) the Jaccard similarity between the 20 nearest neighbors of D1 vs. D2. The first score evaluates the preservation of global structure, while the second metric quantifies the degree of local neighborhood preservation for each cell. We then computed an overall ranking of each t-SNE based on these scores, both of which were averaged across cell types:

We found that the most optimal t-SNE (green) had a better ranking compared to the original t-SNE included in our preprint (red). Although our optimal t-SNE doesn't have the highest Spearman correlation or Jaccard similarity, its scores represent a maximized compromise between preserving local neighborhood integrity, and retaining global structure:

Our new t-SNE shows clear separation of cell type neurotransmitter identities and main brain regions. We also strongly emphasize that our t-SNE is used for the sole purpose of conveying the quality of our snRNA-seq data. We are not using these embeddings to cluster our data, identify cell types, or perform other downstream analyses. The resulting t-SNE is reproduced below:

Referee #2:

Langlieb and Sachdev et al. present a comprehensive atlas of the mouse brain, using single-nucleus RNA sequencing and spatial transcriptomics. The combination of snRNA-seq and Slide-seq is stunning and will be a fantastic resource for mouse brain research. Having the spatial data is extremely important, as it potentially helps interpret the much more abstract single-cell data. It also helps align the molecular cell type atlas with past and future imaging data.

We appreciate the feedback on our datasets.

The technical aspects are excellent and as a resource, the results will be invaluable. I have only minor questions and requests (see below). The biological analyses were fine for what they were, with some caveats (below). For example, the discovery of combinatorial markers of cell types will be a valuable resource for experimentalists to access defined cell types, but the discussion of those markers was perhaps unnecessarily extensive as they provided little insight into the brain's organisation.

Primarily, we view this work as a community resource, and were excited that our algorithm could deliver a clear path towards the principled design of genetic tools to target cell types. Nonetheless, we agree that our discussion of combinatorial markers was too extensive and overly technical. We have therefore moved these analyses to the extended data figures and significantly truncated their discussion in the manuscript, making room for the analyses described in response to the reviewer's other major comment below.

What I found lacking however, was any kind of discussion of organising principles. One gets the impression from the current manuscript that the brain is just a bag of 4000+ neuron types and tens of glial types. But having a combined single-cell and spatial dataset is a great opportunity to discover how the brain is organised. For example, it would probably be useful to define one or more intermediate levels of analysis, e.g. “meta-clusters” or clades in a dendrogram. That may reveal broader patterns of neural cell types, e.g. beyond telencephalic vs non-telencephalic astrocytes, or the distribution of telencephalic excitatory neurons outside the cortex. It may also reveal classes of cells that migrate during development, e.g. ganglionic eminence-derived cortical interneurons, and their possible migration to other target structures e.g. thalamus.

We appreciate this feedback, and recognize that the interpretation and utilization of our dataset would be enabled by a more structured understanding of the cell types we identified in the brain.

First, in response to the reviewer’s suggestion, we generated a dendrogram of our clusters using the Paris hierarchical clustering algorithm⁸; a major challenge with such an enormous dendrogram is the arrangement of the leaves in an order that facilitates the identification of biologically interpretable groups. To address this, we selected a set of discrete, biologically interpretable markers—mostly transcription factors (TFs)—that can be used to assign cell types to embryonic lineages. Then we ran an iterative optimization procedure to reorder the tree’s nodes to maximize the extent to which the lineage-defining TFs are expressed contiguously. We did so by first constructing a normalized gene-by-cell-type matrix, and permuting the gene and cluster order to diagonalize the matrix, using the slanter R package⁹. Then, using a dynamic programming algorithm¹⁰, we optimized the matching between our tree-constrained clusters and this optimized cluster order. Note that this reordering does not change the underlying dendrogram structure, only permuting the nodes’ plotting order for visualization.

Using this ordered tree, and the spatial localizations of each cluster, we aggregated the clusters into a collection of 224 larger meta-clusters, as suggested by the reviewer. By combining key marker expression with spatial localization, we could assign many of these meta-clusters interpretable names, such as “Pallidal cholinergic neurons” (meta-clusters #12) or “MGE-derived PV isocortical interneurons” (meta-cluster #24). Attached in supplementary table 5 is the complete meta-cluster assignment, with differentially expressed genes per meta-cluster in supplementary table 6. Overall, because of the strong separation of these meta-clusters by embryonically specified TF patterns, we conclude that the gene expression dendrogram of neurons largely recapitulates their embryonic specification in development.

We expect that the annotation of cell clusters into meta-clusters with the accompanying dendrogram will make it easier for neuroscientists to interact with, and extract meaning from, these data.

More broadly, are there any organising principles of brain cell types? Do they reflect development? Evolution? function (e.g. neurotransmitter)? Something else? Are glia and neurons organised according to similar patterns, or not? Were brain cell types organised hierarchically? Combinatorially? Was the organising principle different in e.g. telencephalon vs brainstem? What was the relationship between cortical areas? Do any cortical areas stand out with having unique cell types?

The spatial images shown reveal stunningly precise localisation of some neuron types, but I would have loved a more systematic analysis.

For example, the brainstem is traditionally considered to consist of "nuclei" - does this hold true? E.g. are most brain stem neuron types confined to precisely localised nuclei, or are they more dispersed? Are there different classes of neurons - layered (as in cortex), condensed (nuclei), dispersed, ...? If so, do the classes have anything in common? Integrating spatial and single-cell data presents an opportunity of unprecedented analysis of cortical neurons. Could you computationally flatten the cortical sheet and ask about the organising principles of neurons across the sheet? There is ample evidence for e.g. spatial gene expression gradients and differences in cell type composition between cortical areas, but you are in a position to almost have the final word. How are cortical neurons organised spatially across the cortical sheet? Are there systematic groups of such cell types? Are the layer-specific neurons organised according to the same or different principles? Do neocortical types extend into piriform or entorhinal cortex, or the amygdala? If so, which ones do and which ones don't. Etc.

The reviewer has many interesting and intriguing ideas here, and we greatly appreciate the suggestions and encouragement to go deeper with our analyses. In our generation of this resource, our utmost priority has been in delivering a reference dataset that the community can trust, and leverage for many basic applications in neuroscience. We freely admit this has taken an incredible amount of time and effort, such that the extraction of biological principles was hard to prioritize.

Since our initial submission, we have focused our analytical attention on answering the following question: for a given neuronal cell type, how do its closest neighbors distribute across different brain areas? We wanted to understand whether certain kinds of neurons might be more or less likely to have close molecular relatives in other brain regions.

Using the dendrogram we generated (described above), for each cell type, we defined a cellular neighborhood, based upon consecutively aggregating descendants from successively more distant ancestors in the tree from that index cell cluster. We calibrated this distance threshold both by the number of clusters that compose the neighborhood, and by ensuring that the clusters included did not cross major cell type definitional boundaries (e.g. ensuring that neurons and glia do not mix within a neighborhood). Using this definition, we then asked how a cluster's neighborhood is distributed across the brain. Below are density plots of the regional localization of all clusters composing a cell type's neighborhood (each thin line is one cell type). The cell types are grouped together by whether they are projection neurons (neurons that send long-range axons outside of the structure) or interneurons (neurons that signal locally) (Fig. 2c):

Examining these results for cell types in the isocortex, striatum, and cerebellum, we find that regional specialization in the brain is strongest in the principal, long-range-projecting neurons of individual structures, while interneurons are more likely to retain molecular features of cell types found in more distinct and evolutionarily distant brain areas. Our isocortex analysis is consistent with decades of fate mapping results showing that medial and caudal ganglionic eminence-derived interneurons are widespread throughout the telencephalon. In addition, we find molecularly similar cell clusters—expressing *Lhx6*, *Dlx1/2*, *Arx*, and *Nkx2-1*—within the hypothalamus. *Lhx6* is required for hypothalamic inhibitory neuron specification¹¹, and a recent study of adult human brain also identified inhibitory populations within the hypothalamus that are molecularly related to cortical inhibitory neurons¹.

We find it interesting that the relative retention of similar interneuron identities across more diverse brain structures is not exclusive to the telencephalon. In particular, we identified cell types molecularly similar to both excitatory and inhibitory cerebellar interneurons (but not Purkinje projection cells) within other brainstem structures, most especially the dorsal and ventral cochlear nuclei (Fig. 2d, reproduced below):

We therefore hypothesize that molecular specialization proceeds more rapidly within the projection neurons of a structure, compared with the interneurons. Our analysis was restricted to regions where we can definitively connect transcriptionally defined clusters to interneuron, or projection neuron, identities. It will be interesting to pressure-test this hypothesis in other regions, once more comprehensive maps that combine neuronal projections and transcriptional profiling become available (for example, by using viral barcoding approaches).

Another interesting aspect concerns sex. Were you powered to study sex differences, and did you find any? Were there sex differences in cell types, in cell type proportions, or in specific genes expressed in cell types?

One of the most sexually dimorphic regions of the brain is the bed nucleus of the stria terminalis (BNST). Early in the project, to develop an understanding of the minimum replicate power we would need to discover dimorphism, we collected many animal replicates from BNST (7 female and 8 male, total of 56,766 neuronal nuclei profiles, see Welch et al¹² for details of our results with these data). Using co-varying neighborhood analysis (CNA¹³), we tested for significant association of cell abundances in relation to sample identity (male or female). We find that we are barely powered by our 15 mouse samples to obtain a significant global association ($p < 0.05$). Individual cell neighborhoods reveal distinct populations of neurons, many of which are negatively associated with sex (i.e. more associated with male samples) at FDR < 0.01 . Here we plot a t-SNE of just the BNST cells highlighted by their global association p-value:

Here are the number of male and female replicates we were able to collect across dissections:

If we downsample to only six replicates of 3 male and 3 female donors (closer to the average number of replicates per dissectate), our CNA results do not reveal any sex-biased neighborhoods, as none were identified with statistical confidence ($p < 0.05$), and no global deviation in this region with well-known dimorphism:

We therefore conclude that substantially larger replicate numbers per region would be needed to address the dimorphism question than we were resourced to collect for this project.

* Technical issues

On p. 10 “Regional dissections”: what were the precise extents of the anatomical regions, and how confident were the borders of the dissected tissue blocks? In other words, to what extent should we expect a region to also contain contaminating cells from adjacent regions, or to potentially miss cells located near its borders? For example, the photo in Fig 1a suggests that dissections deviated quite strongly from neuroanatomical borders. If the photo documentation is available, it would be great if the authors would share it e.g. on a companion web site. Alternatively, polygon coordinates could be added to the supplementary table that defines dissections, so that they can be overlaid on the anatomical coordinate framework.

As part of our data preparation, our neuroanatomist, who led the construction of the single- nuclei dataset generation, carefully annotated each of the Slide-seq arrays with the corresponding dissection regions. As we align the Slide-seq arrays into a cohesive 3-D space annotated with the Allen Institute’s CCF regions, these dissection annotations are also then integrated into CCF coordinate space. We make these dissection annotations available on our accompanying site, where we have annotated each bead with all the corresponding metadata, including CCF XYZ coordinates, dissection ID’s, and nissl stain XY coordinates.

The tSNEs in Fig 1 use parameters that lead to filling the unit circle, again giving the impression of a bag of cell types. It would probably be informative to increase the exaggeration parameter so as to yield a stronger global structure (this can be done e.g. with OpenTSNE; see also the paper “The art of using tSNE for single-cell transcriptomics”, which has good heuristics for the parameters). UMAP will typically have a similar effect. Perhaps this would help define additional intermediate levels of analysis mentioned above.

We appreciate the reviewer’s helpful suggestions on how to improve our t-SNE visualization. Using OpenTSNE⁶, we sought to find optimal t-SNE settings that maximized both local and global structure, based on the guidelines laid out by the authors of “The art of using t-SNE for single-cell transcriptomics”⁷. Namely, we performed a sweep of parameters, varying both the perplexity and exaggeration factor of our t-SNE coordinates. To determine which t-SNE settings were most optimal, we compared the pairwise euclidean distances of cell type t-SNE centroids (D1), to the pairwise euclidean distances of cell type %expression across highly variable genes (D2). We computed two scores: 1) the Spearman correlation of D1 vs. D2, and 2) the Jaccard similarity between the 20 nearest neighbors of D1 vs. D2. The first score evaluates the preservation of global structure, while the second metric quantifies the degree of local neighborhood preservation for each cell. We then computed an overall ranking of each t-SNE based on these scores, both of which were averaged across cell types:

We found that the most optimal t-SNE (green) had a better ranking compared to the original t-SNE included in our preprint (red). Although our optimal t-SNE doesn't have the highest Spearman correlation or Jaccard similarity, its scores represent a maximized compromise between preserving local neighborhood integrity, and retaining global structure:

Our new t-SNE shows clear separation of cell type neurotransmitter identities and main brain regions. We also strongly emphasize that our t-SNE is used for the sole purpose of conveying the quality of our snRNA-seq data. We are not, however, using these embeddings to cluster our data, identify cell types, or perform other downstream analyses.

For Extended Fig 1g, a table of numbers would be more informative (log-scaled bars are very hard to read anyway). Were there exactly 1 microglia and 1 OPC cluster, i.e. those zero-height bars are actually informative?

We thank the reviewer for suggesting ways to improve the clarity of our cell class tables in this supplementary figure. We have replaced Extended Data Fig. 1h with log-scaled bars of nuclei across cell class. The raw number of clusters and nuclei across cell class are also available in Supplemental Table 3.

** Improved analysis of non-neuronal cell types.*

We appreciate both reviewers' general encouragement to pursue a more detailed annotation and analysis of the glial populations. We admit that our fascination with the extraordinary diversity in the neurons occupied the great majority of our attention, and we somewhat neglected the glia in our initial submission.

Below the reviewer can find specific responses to each question, which are now jointly included in the manuscript in Extended Data Figure 2d. In addition, we have added a small section to the beginning of the results section of the main text about our findings, reproduced below:

"Most glial populations were distributed across large neuroanatomical boundaries (telencephalon, mesencephalon, and rhombencephalon), indicating that, relative to neurons, regional gene expression differences amongst glial populations were small (Extended Data Fig. 2d). A single oligodendrocyte precursor (OPC) cluster was identified, in contrast to a recent report of additional OPC subspecialization in humans¹. The glial clusters with regional segmentation included astrocytes, which divided into olfactory-specific, telencephalic, and non-telencephalic populations, as well as a cerebellum-specific population (the Bergmann glia). Amongst our endothelial cell populations, we identified populations preferentially localized to the choroid plexus (Extended Data Fig. 2e). Additional regionally localized glial populations included the olfactory ensheathing neurons, identified by their expression of the known marker homeobox genes *A1x3* and *A1x4*², and hypothalamic tanycytes, which uniquely express *Rax3*³."

Were there ~20 endothelial cell types? What were they? Did you identify any choroid plexus-specific endothelial cells, with possible specialised function in producing CSF fluid? To my knowledge, previous work in mouse has failed to identify convincing region-specific endothelial cells. Did you find any? Or do all the ~20 clusters correspond to the arterial-capillary-venous gradient?

Utilizing markers from two recent papers^{14,15}, we were able to annotate our 22 endothelial types in three major groups: pericytes (n= 5), vascular smooth muscle cells (VSMCs, n=4), and proper endothelial cells (ECs, n=13). One VSMC type appeared to be contractile, as judged by its expression of *Actg2*. The ECs could further be divided into arterial (n=4), capillary (n=3), and venous (n=6) groups.

Interestingly, perhaps as the reviewer suggested, amongst these 22 types, we identified some populations that were more specific to the choroid plexus. We confirmed this in the Slide-seq localization of one of these populations, Endo-Flt1-Exoc3l2-1 (figure reproduced from our web resource, braincelldata.org):

In addition, two pericyte populations—Endo_Flt1_Kcnj8_Slc47a2_1 and Endo_Flt1_Kcnj8_Slc47a2_2—specifically expressed *Steap4*, which has very clear choroid localization both in our Slide-seq atlas (left) and in the Allen Institute ISH atlas (right):

Aside from these very interesting choroid populations we uncovered with the reviewer's help, we did not identify any other endothelial types that were specific to any brain structures.

In our revised manuscript, we include a sentence about these choroid-specific endothelial populations (reproduced above). In addition, we have included more detailed endothelial cluster annotations as part of the metadata available on our portal, braincelldata.org.

Extended Fig 2d is hard to read since there is no annotation on the vertical axis. There seem to be >50 glial types, but what are they? Please indicate what they are (ependymal, astrocyte, ...). There seems to be quite a lot of region-specific glia. What were they?

We apologize for not properly labeling and annotating the heat map in Extended Fig 2d, as we fear it may have falsely elevated the reviewer’s excitement about region-specific glia. In fact, all of the region specificities are as expected, based upon known cell type localizations. Below is an updated version of this plot, which includes annotation of the rows, showing the major groups of glia:

As shown, we identified clusters of telencephalic versus non-telencephalic astrocytes, and specialized astrocyte populations in cerebellum (Bergmann glia) and olfactory bulb. In endothelial cells, we found some clusters with localizations similar to ependymal cells (these correspond to the choroid-enriched populations described above). Some fibroblast populations also appeared to be more localized to the ventricles. The remaining spatially localized populations were olfactory ensheathing neurons and tanycytes. Regional specializations of each of these cell types has been described already, except for perhaps the endothelial cells described in the reply above.

We have revised the Extended Data Fig. 2d to include this information, and included additional discussion of glial cell types in the main text of the manuscript, as reproduced above.

Did you find only one kind of OPC? That would agree with previous work in mice, but disagree with recent work in the human brain (e.g. Siletti et al. bioRxiv 2022). It would be potential evidence of a potential human vs mouse specialisation!

Indeed, we only identified a single OPC cluster in the mouse brain. In our revision, we cite Siletti et al. in describing the potential contrast between the species.

You seem to find multiple types of macrophages - what were they? E.g. distinct macrophages of different brain borders (meninges, vascular, ...)?

We identified three distinct populations of macrophages. One expresses *Cd209a* and *Slamf7*, suggesting some dendritic cell-like properties. Another is positive for *Cd163*, indicating it is likely an M2 macrophage population. The third population expresses *H2-aa*, *C1qb*, and *Klra2*, possibly indicating a more M1-like state.

Each of these populations lacks a clear, interpretable spatial localization—they are merely found randomly and sparsely scattered throughout the brain parenchyma.

We have updated the portal site to include more annotation information about these macrophage populations.

You seem to find a large number of fibroblasts types — interesting! What were they? Meninges layers? Did you remove meninges, or not (please add to Methods)? If you have meninges, do you see any evidence of the recently described SLYM layer? It has been very controversial. Were the meninges different dorsally vs ventrally, or anterior vs posterior? This is relevant because meningioma subtypes have characteristic spatial patterns. Did you find specialised perivascular fibroblasts?

Prior to dissection for snRNA-seq and sectioning for Slide-seq, we attempted to remove the meninges from the entire surface of the brain (and have now updated our Methods section to note this). However, while removal of the dura was easy to visualize, we strongly suspect that pia and some arachnoid material remained.

In response to the reviewer's suggestion, we attempted to more carefully annotate our fibroblast populations based upon results from two studies of fibroblast diversity and development in the brain^{16,17}:

Many fibroblast clusters ($n=11$) expressed markers of a parenchymal perivascular fibroblast identity¹⁶, including *Lum*, *Col3a1*, *Pdgfra*, and *Dcn*, and were negative for expression of *Mgp* and *Slc4a4*. Another 21 clusters had high expression of *Mgp* and/or *Slc4a4*, and had low or absent expression of perivascular markers, likely indicating a meningeal origin. The remaining three clusters lacked markers of fibroblasts previously studied in the brain, but their expression of *Mmp2* and *Mmp13* might indicate stromal-like identity. Amongst the putative meningeal fibroblasts, all expressed *Lama2* and some expressed *Aldh1a2*, indicating they are likely to be localized to the pia or arachnoid¹⁶.

We have updated the annotation information on our portal to reflect these new, more detailed annotations.

To ask whether any of the putative meningeal types correspond to the SLYM layer discussed in Møllgård et al 2023¹⁸, we plotted a series of markers described in that paper that the authors indicated distinguish these cells from those in the arachnoid and pia: *Prox1*, *Pdpm*, *Lyve1*, *Crabp2*, *Flt4*, *Cldn11*, and *Cdh1*:

The reported SLYM profile is: *Prox1*+/*Pdpn*+/*Lyve1*-/*Crabp2*+/*Flt4*-/*Cldn11*-/*Cdh1*-. While expression of many of these genes is low (bear in mind ours is an snRNA-seq dataset, compared with the immunostaining primarily done in Møllgård *et al*), we do not think any cluster matches the SLYM profile (note especially the very strong correlation between *Prox1* and *Cdh1*)— but we caution that these results are hardly definitive.

Were there no other immune cells? That in itself would be very interesting, since there is a common belief that there must be some lymphoid cells in the normal brain even if they are rare. You could put a number on that.

Although these cells were excluded from most analyses, they have been included in the following: details about our QC strategy (Extended Data Fig. 1a, Supplemental Table 2), distribution of nUMIs across cell class (Extended Data Fig. 1d) and number of nuclei across cell class (Extended Data Fig. 1h, Supplemental Table 3), where they are clearly shown to be an extremely rare population. We have also now, in our revision, included them in the new Extended Data Fig. 2d. Given their rarity, and our inability to visualize them at very high spatial resolution, we cannot determine which are truly parenchymal and which are lingering within vascular compartments.

Citations

1. Siletti, K. *et al.* Transcriptomic diversity of cell types across the adult human brain. *bioRxiv* 2022.10.12.511898 (2022) doi:10.1101/2022.10.12.511898.
2. Perera, S. N. *et al.* Insights into olfactory ensheathing cell development from a laser-microdissection and transcriptome-profiling approach. *Glia* **68**, 2550–2584 (2020).
3. Miranda-Angulo, A. L., Byerly, M. S., Mesa, J., Wang, H. & Blackshaw, S. Rax regulates hypothalamic tanycyte differentiation and barrier function in mice. *J. Comp. Neurol.* **522**, 876–899 (2014).
4. Wang, Q. *et al.* The Allen Mouse Brain Common Coordinate Framework: A 3D reference atlas. *Cell* **181**, 936–953.e20 (2020).
5. Dasen, J. S. Chapter 4 Transcriptional Networks in the Early Development of Sensory–Motor Circuits. in *Current Topics in Developmental Biology* vol. 87 119–148 (Academic Press, 2009).
6. Poličar, P. G., Stražar, M. & Zupan, B. openTSNE: a modular Python library for t-SNE dimensionality reduction and embedding. *bioRxiv* 731877 (2019) doi:10.1101/731877.
7. Kobak, D. & Berens, P. The art of using t-SNE for single-cell transcriptomics. *Nat. Commun.* **10**, 5416 (2019).
8. Bonald, T., Charpentier, B., Galland, A. & Hollocou, A. Hierarchical Graph Clustering using Node Pair Sampling. (2018).
9. Meir, Z. *et al.* Dissection of floral transition by single-meristem transcriptomes at high temporal resolution. *Nat Plants* **7**, 800–813 (2021).
10. Venkatachalam, B., Apple, J., St John, K. & Gusfield, D. Untangling tanglegams: comparing trees by their drawings. *IEEE/ACM Trans. Comput. Biol. Bioinform.* **7**, 588–597 (2010).
11. Kim, D. W. *et al.* Gene regulatory networks controlling differentiation, survival, and diversification of hypothalamic Lhx6-expressing GABAergic neurons. *Commun Biol* **4**, 95 (2021).
12. Welch, J. D. *et al.* Single-Cell Multi-omic Integration Compares and Contrasts Features of Brain Cell Identity. *Cell* (2019) doi:10.1016/j.cell.2019.05.006.
13. Reshef, Y. A. *et al.* Co-varying neighborhood analysis identifies cell populations associated with phenotypes of interest from single-cell transcriptomics. *Nat. Biotechnol.* **40**, 355–363 (2022).
14. Wakabayashi, T. & Naito, H. Cellular heterogeneity and stem cells of vascular endothelial cells in blood vessel formation and homeostasis: Insights from single-cell RNA sequencing. *Front Cell Dev Biol* **11**, 1146399 (2023).
15. Dani, N. *et al.* A cellular and spatial map of the choroid plexus across brain ventricles and ages. *Cell* **184**, 3056–3074.e21 (2021).
16. Vanlandewijck, M. *et al.* A molecular atlas of cell types and zonation in the brain vasculature. *Nature* **554**, 475–480 (2018).
17. DeSisto, J. *et al.* Single-Cell Transcriptomic Analyses of the Developing Meninges Reveal Meningeal Fibroblast Diversity and Function. *Dev. Cell* **54**, 43–59.e4 (2020).
18. Møllgård, K. *et al.* A mesothelium divides the subarachnoid space into functional compartments. *Science* **379**, 84–88 (2023).

Reviewer Reports on the First Revision:

Referees' comments:

Referee #1 (Remarks to the Author):

The authors have directly addressed every one of my comments and suggestions thoroughly. The manuscript was already strong and I believe the revisions have taken it even further and improved clarity and accessibility. This work is exceptional and will be a powerful reference for many studies to come. I look forward to seeing it published.

-Andrew Adey

Referee #2 (Remarks to the Author):

The authors have done a thorough job of responding in detail to all my (many) questions and suggestions. Their paper will be a beautiful and rich resource that will serve the neuroscience community for years to come. I have nothing further, and wish to congratulate the authors on their landmark paper.